# Climate and air quality impacts due to mitigation of non-methane near-term climate forcers

Robert J. Allen[1], Steven Turnock[2], Pierre Nabat[3], David Neubauer[4], Ulrike Lohmann[4], Dirk Olivié[5], Naga Oshima[6], Martine Michou[3], Tongwen Wu[7], Jie Zhang[7], Toshihiko Takemura[8], Michael Schulz[5], Kostas Tsigaridis[9], Susanne E. Bauer[9], Louisa Emmons[10], Larry Horowitz[11], Vaishali Naik[11], Twan van Noije[12], Tommi Bergman[12,13], Jean-Francois Lamarque[14], Prodromos Zanis[15], Ina Tegen[16], Daniel M. Westervelt[17], Phillipe Le Sager[12], Peter Good[2], Sungbo Shim[18], Fiona O'Connor[2], Dimitris Akritidis[15], Aristeidis K. Georgoulias[15], Makoto Deushi[6], Lori T. Sentman[11], Jasmin G. John[11], Shinichiro Fujimori[19,20,21], and William J. Collins[22]

[1]Department of Earth and Planetary Sciences, University of California Riverside, Riverside, CA, 92521 USA
[2]Met Office Hadley Centre, Exeter, UK
[3]Centre National de Recherches Meteorologiques (CNRM), Universite de Toulouse, Meteo-France, CNRS, Toulouse, France
[4]Institute of Atmospheric and Climate Science, ETH Zurich, Zurich, Switzerland
[5]Norwegian Meteorological Institute, Oslo, Norway
[6]Meteorological Research Institute, Japan Meteorological Agency
[7]Beijing Climate Center, China Meteorological Administration, Beijing, China
[8]Research Institute for Applied Mechanics, Kyushu University, Fukuoka, Japan
[9]Center for Climate Systems Research, Columbia University, NASA Goddard Institute for Space Studies, USA
[10]Atmospheric Chemistry Observations and Modelling Lab, National Center for Atmospheric Research, Boulder, CO, USA
[11]DOC/NOAA/OAR/Geophysical Fluid Dynamics Laboratory. Biogeochemistry, Atmospheric Chemistry, and Ecology 10 Division, Princeton, USA
[12]Royal Netherlands Meteorological Institute, De Bilt, Netherlands
[13]Finnish Meteorological Institute, Helsinki, Finland
[14]NCAR/UCAR, Boulder, CO, USA
[15]Department of Meteorology and Climatology, School of Geology, Aristotle University of Thessaloniki, Thessaloniki, Greece
[16]Leibniz Institute for Tropospheric Research, Leipzig, Germany
[17]Lamont-Doherty Earth Observatory, Columbia University, Palisades, New York, USA
[18]National Institute of Meteorological Sciences, Seogwipo-si, Jeju-do, Korea
[19]Department of Environmental Engineering, Kyoto University, C1-3 361, Kyotodaigaku Katsura, Nishikyoku, Kyoto city, Japan
[20]Center for Social and Environmental Systems Research, National Institute for Environmental Studies (NIES), 16-2 Onogawa, Tsukuba, Ibaraki, 305-8506, Japan
[21]International Institute for Applied System Analysis (IIASA), Schlossplatz 1, A-2361, Laxenburg, Austria
[22]Department of Meteorology, University of Reading, Reading, RG6 6BB, UK

**Correspondence:** Robert J. Allen (rjallen@ucr.edu)

**Abstract.** It is important to understand how future environmental policies will impact both climate change and air pollution. Although targeting near-term climate forcers (NTCFs), defined here as aerosols, tropospheric ozone and precursor gases, should improve air quality, NTCF reductions will also impact climate. Prior assessments of the impact of NTCF mitigation on air quality and climate have been limited. This is related to the idealized nature of some prior studies, simplified treatment

of aerosols and chemically reactive gases, as well as a lack of a sufficiently large number of models to quantify model diversity and robust responses. Here, we quantify the 2015-2055 climate and air quality effects of non-methane NTCFs using nine state-of-the-art chemistry-climate model simulations conducted for the Aerosol and Chemistry Model Intercomparison Project (AerChemMIP). Simulations are driven by two future scenarios featuring similar increases in greenhouse gases (GHGs) but with "weak" (SSP3-7.0) versus "strong" (SSP3-7.0-lowNTCF) levels of air quality control measures. As SSP3-7.0 lacks climate policy and has the highest levels of NTCFs, our results (e.g., surface warming) represent an upper bound. Unsurprisingly, we find significant improvements in air quality under NTCF mitigation (strong versus weak air quality controls). Surface fine particulate matter ($PM_{2.5}$) and ozone ($O_3$) decrease by $-2.2\pm0.32$ $\mu$g m$^{-3}$ and $-4.6\pm0.88$ ppb, respectively (changes quoted here are for the entire 2015-2055 time period; uncertainty represents the 95% confidence interval), over global land surfaces, with larger reductions in some regions including south and southeast Asia. Non-methane NTCF mitigation, however, leads to additional climate change due to the removal of aerosol which causes a net warming effect, including global mean surface temperature and precipitation increases of $0.25\pm0.12$ K and $0.03\pm0.012$ mm day$^{-1}$, respectively. Similarly, increases in extreme weather indices, including the hottest and wettest day, also occur. Regionally, the largest warming and wetting occurs over Asia, including central and north Asia ($0.66\pm0.20$ K and $0.03\pm0.02$ mm day$^{-1}$), south Asia ($0.47\pm0.16$ K and $0.17\pm0.09$ mm day$^{-1}$) and east Asia ($0.46\pm0.20$ K and $0.15\pm0.06$ mm day$^{-1}$). Relatively large warming and wetting of the Arctic also occur at $0.59\pm0.36$ K and $0.04\pm0.02$ mm day$^{-1}$, respectively. Similar surface warming occurs in model simulations with aerosol-only mitigation, implying weak cooling due to ozone reductions. Our findings suggest that future policies that aggressively target non-methane NTCF reductions will improve air quality, but will lead to additional surface warming, particularly in Asia and the Arctic. Policies that address other NTCFs including methane, as well as carbon dioxide emissions, must also be adopted to meet climate mitigation goals.

## 1   Introduction

Near-term climate forcers (NTCFs), also referred to as short-lived climate forcers (SLCFs), are those chemical species whose impact on climate occurs primarily within the first decade after their emission (Myhre et al., 2013). This set of compounds includes ozone, aerosols, and their precursor gases, as well methane ($CH_4$) which is also a well-mixed greenhouse gas (GHG). Other well-mixed GHGs, including carbon dioxide ($CO_2$) and nitrous oxide ($N_2O$), possess much longer atmospheric lifetimes and impact climate on decadal to centennial time scales.

NTCFs have important impacts on the climate system and human health, as they perturb the radiative balance of Earth and contribute to air pollution. The total aerosol radiative effect, estimated as an effective radiative forcing (ERF), is $-0.9$ W m$^{-2}$ with a 90% confidence range of $-1.9$ to $-0.1$ W m$^{-2}$ (Boucher et al., 2013). A more recent review revised the 90% confidence range to more negative values ($-2.0$ to $-0.4$ W m$^{-2}$) (Bellouin et al., 2020). Moreover, not all aerosols have a negative forcing, as black carbon (BC) from anthropogenic fossil and biofuel emissions possesses a radiative forcing of $+0.40$ (0.05 to 0.80) W m$^{-2}$. BC, however, is often associated with co-emission of organic matter. The best estimate of net industrial-era climate forcing by all short-lived species from black-carbon-rich sources, including open burning emissions, is slightly negative

but with relatively large uncertainty bounds of $-1.45$ to $+1.29$ W m$^{-2}$ (Bond et al., 2013). Thus, changes in BC emissions that are different from changes in non-absorbing aerosols will lead to differing ERF changes. Tropospheric ozone, which is formed in the atmosphere through chemical reactions between nitrogen oxides (NO$_x$), carbon monoxide (CO), and volatile organic compounds (VOCs) including methane in the presence of sunlight, also exhibits a positive forcing of $+0.40\pm0.2$ W m$^{-2}$ (Myhre et al., 2013). The radiative forcing of changes in methane concentrations is estimated at $0.48\pm0.05$ W m$^{-2}$ (Myhre et al., 2013). We note that these estimates are currently being updated as part of the Coupled Model Intercomparison Project version 6 (CMIP6) (Pincus et al., 2016; Eyring et al., 2016). Thus, reductions in some NTCFs, including non-absorbing aerosols, will warm the climate system, whereas reductions in other NTCFs, including absorbing aerosols, tropospheric ozone, and methane will cool the climate system. Things become more complex from an emissions perspective, as reductions in some precursor gases such as NO$_x$ and VOCs impact ozone, methane and aerosols (Myhre et al., 2013). Reductions in NO$_x$, for example, will promote cooling due to reduced tropospheric ozone, but the impact on CH$_4$ lifetime and aerosol formation may promote overall warming (Fiore et al., 2015).

NTCFs also perturb the hydrological cycle. Energetic constraints and modeling studies show that anthropogenic aerosols lead to reduced global mean precipitation (Ramanathan et al., 2001; Wilcox et al., 2013; Samset et al., 2016). Aerosol induced reductions in surface solar radiation will be partially balanced by reductions in latent cooling, leading to corresponding rainfall reductions. In the case of absorbing aerosols−particularly in the boundary layer−atmospheric heating stabilizes the atmosphere and reduces convection, also leading to an overall decrease in precipitation (Ming et al., 2010; Ban-Weiss et al., 2012; Stjern et al., 2017; Allen et al., 2019a; Johnson et al., 2019). The buildup of aerosols during the 20th century has helped mask the expected increase in global mean precipitation due to GHG-induced warming (Liepert et al., 2004; Wu et al., 2013; Salzmann, 2016; Richardson et al., 2018). Furthermore, studies show that the hemispheric contrast in aerosol forcing has shifted the tropical rainbelt southward, which is associated with a weakening of the west African Monsoon and the occurrence of the Sahel drought of the mid-1980s (Rotstayn and Lohmann, 2002; Biasutti and Giannini, 2006; Allen and Sherwood, 2011; Ackerley et al., 2011; Chang et al., 2011; Biasutti, 2013; Hwang et al., 2013; Dong et al., 2014; Allen et al., 2015; Undorf et al., 2018). The observed precipitation decrease during recent decades over most of the areas affected by the South and East Asia monsoon can also be explained by the dominance of aerosol radiative effects suppressing precipitation over the expected precipitation enhancement due to increased GHGs (Wang et al., 2013; Song et al., 2014; Li et al., 2015; Xie et al., 2016; Krishnan et al., 2016; Guo et al., 2016; Lau and Kim, 2017; Zhang et al., 2017; Lin et al., 2018; Liu et al., 2018).

NTCFs are also a source of air pollution, including surface ozone (O$_3$) and fine particulate matter less than 2.5 $\mu$m in diameter (PM$_{2.5}$). Air pollution has negative impacts on human health, including exacerbation of cardiovascular and respiratory diseases, and cancer. Recent estimates show air pollution is the 4th-highest ranking risk factor for premature death and mainly due to non-communicable diseases, responsible for $\sim$7 million premature deaths per year, with 4.2 million of these annual deaths attributable to ambient air pollution (WHO, 2016; Cohen et al., 2017; Butt et al., 2017). A more recent study suggests the global total excess mortality rate due to all air pollution is 8.79 million per year (95% confidence interval of 7.11-10.41 million per year), leading to a global mean loss of life expectancy of 2.9 years (Lelieveld et al., 2019).

Future reductions in emissions of NTCFs are necessary for improved air quality, but will yield relatively rapid (i.e., decadal) climate responses due to their short atmospheric lifetimes (relative to GHGs). Samset et al. (2018) show that complete removal of present-day anthropogenic aerosol emissions induces a global mean surface heating of 0.5-1.1K and a precipitation increase of 2-4.6%. Similar large, near-term increases in global warming and precipitation are predicted by other studies that assume a rapid removal of anthropogenic aerosols (Brasseur and Roeckner, 2005; Andreae et al., 2005; Ramanathan and Feng, 2008; Raes and Seinfeld, 2009; Kloster et al., 2010; Arneth et al., 2009; Matthews and Zickfeld, 2012; Rotstayn et al., 2013; Wu et al., 2013; Westervelt et al., 2015; Salzmann, 2016; Hienola et al., 2018; Richardson et al., 2018; Lelieveld et al., 2019). Furthermore, future aerosol reductions may shift the tropical rainbelt northward and may strengthen precipitation in several monsoon regions, including West Africa, South Asia, and East Asia (Levy et al., 2013; Allen, 2015; Rotstayn et al., 2015; Allen and Ajoku, 2016; Westervelt et al., 2017; Zhao et al., 2018; Westervelt et al., 2018; Scannell et al., 2019; Zanis et al., 2020). In contrast to the above studies, however, Shindell and Smith (2019) show that the time required to transform power generation, industry and transportation leads to largely offsetting climate impacts of $CO_2$ and sulfur dioxide (a precursor of sulfate aerosol), implying no conflict between climate and air-quality objectives. Their simulations use a simple emissions-based climate model, Finite Amplitude Impulse Response (FAIR) (Smith et al., 2018), and it is not known if this result also applies to fully coupled chemistry-climate models.

Despite the rich literature, a robust assessment of the impact of specific NTCF mitigation measures on climate and air quality has been difficult to achieve. Part of this uncertainty stems from the idealized nature of many of the prior studies (e.g., instantaneous removal of all aerosols), simplified treatment of aerosols and chemically reactive gases, as well as a lack of a sufficiently large number of models performing identical simulations with which to quantify model diversity and robust responses. The Aerosol and Chemistry Model Intercomparison Project (AerChemMIP) (Collins et al., 2017), part of CMIP6 (Eyring et al., 2016), quantifies the climate and air quality impacts of aerosols and chemically reactive gases. Here, we use AerChemMIP and the Scenario Model Intercomparison Project (ScenarioMIP, O'Neill et al., 2016) to quantify the climate and air quality impacts due to non-methane NTCF mitigation (aerosols and ozone only) through analysis of two future emission scenarios−one with weak (SSP3-7.0) and one with strong (SSP3-7.0-lowNTCF) levels of air quality control measures. NTCF mitigation is defined here as the difference between these two scenarios, SSP3-7.0-lowNTCF − SSP3-7.0. Models include an interactive representation of tropospheric aerosols and atmospheric chemistry, allowing for the quantification of chemistry-climate interactions. We use mean surface temperature and precipitation, as well as three climate extreme metrics including the hottest and wettest day, and consecutive dry days, as indicators of climate change and surface $O_3$ and $PM_{2.5}$ for air quality as these are commonly used metrics. We show that non-methane NTCF reductions improve air quality, but also lead to additional climate change including surface warming. Policies that address other NTCFs including $CH_4$, as well as $CO_2$ emissions, must also be undertaken. Methods are presented in Section 2 and results are discussed in Section 3. Conclusions appear in Section 4.

## 2 Methods

### 2.1 AerChemMIP Models

Nine coupled ocean-atmosphere climate models performed the SSP3-7.0 and SSP3-7.0-lowNTCF simulations, including CNRM-ESM2-1 (Séférian et al., 2019; Michou et al., 2019), MIROC6 (Takemura et al., 2005, 2009; Tatebe et al., 2019), MPI-ESM1-2-HAM (Mauritsen et al., 2019; Neubauer et al., 2019; Tegen et al., 2019), NorESM2-LM (Seland et al., 2020), BCC-ESM1 (Wu et al., 2019, 2020), GFDL-ESM4 (John et al., 2018; Horowitz et al., 2018; Dunne et al., submitted; Horowitz et al., submitted), CESM2-WACCM (Emmons et al., 2020; Gettelman et al., 2019; Tilmes et al., 2019), UKESM1-0-LL (Sellar et al., 2019) and MRI-ESM2-0 (Yukimoto et al., 2019). However, the first four models (CNRM-ESM2-1, MIROC6, MPI-ESM1-2-HAM, NorESM2-LM) lack interactive tropospheric chemistry schemes and therefore include identical ozone evolution in both SSP3-7.0 and SSP3-7.0-lowNTCF simulations (as recommended by AerChemMIP). As NTCF mitigation only includes the effects of aerosols in these four models, we refer to these models as "Aer". The remaining five models, including BCC-ESM1, GFDL-ESM4, CESM2-WACCM, UKESM1-0-LL and MRI-ESM2-0, include interactive atmospheric chemistry and aerosols, and therefore both aerosol and ozone reductions are included. These models are referred to as "Aer+O3".

In addition to coupled simulations, models also performed analogous fixed-SST experiments to quantify the effective radiative forcing (ERF). The ERF is calculated from the top-of-the-atmosphere (TOA) flux differences between atmosphere-only simulations with identical SSTs but differing composition (Forster et al., 2016; Pincus et al., 2016). The above scenarios (SSP3-7.0 and SSP3-7.0-lowNTCF) are repeated with prescribed SSTs. These SSTs (and sea ice) are taken from the monthly mean evolving values from one ensemble member of the coupled SSP3-7.0 ScenarioMIP run (Collins et al., 2017). MPI-ESM1-2-HAM used daily mean SST and sea ice. The differences in radiative fluxes between the weak and strong air quality control scenarios yield the TOA transient ERF due to NTCF mitigation.

### 2.2 Model Data and Methodology

All models performed at least one realization each of SSP3-7.0 and SSP3-7.0-lowNTCF. CNRM-ESM2-1, MIROC6, UKESM1-0-LL, NorESM2-LM, CESM2-WACCM and BCC-ESM1 performed three realizations of each experiment. For these models, the model mean response (average over the three realizations) is shown. The multi-model mean (MMM) is obtained by averaging each model's mean response (i.e., each model has the same weight). Only one realization exists for the corresponding fixed-SST experiments. Unless otherwise mentioned, all analyses are based on annual means. All data is spatially interpolated to a 2.5°x2.5° grid using bilinear interpolation.

Model trends are calculated using least-squares regression, and the corresponding trend significance is based on a two-tailed Student's $t$-test, where the null hypothesis of a zero regression slope is evaluated. Multi-model mean trends and their significance are calculated using two different methods. In the first method, MMM trends are calculated from the multi-model mean time series using a weighted least-squares regression, where each value in the multi-model mean time series is weighted by $1/\sigma_m^2$, where $\sigma_m$ is the standard deviation across models. We note that the MMM trends and significance are very similar

with and without weighting the regression. Autocorrelation of the time series is also accounted for by using the effective sample size, defined as $n(1 - r_1)/(1 + r_1)$, where $n$ is the number of years and $r_1$ is the lag-1 autocorrelation coefficient.

We also quantify the significance of the multi-model mean trend relative to each individual model mean trend. Here, the MMM trend is calculated as the average of the individual model mean trends and its uncertainty is calculated as plus/minus twice the standard error (i.e., the 95% confidence interval), which is $2\sigma/\sqrt{n_m}$, where $\sigma$ is the standard deviation of the trends and $n_m$ is the number of models. If this confidence interval does not include zero, then the multi-model mean trend is significant at the 95% confidence level. Both methods yield similar conclusions as to the magnitude and significance of the MMM trends.

We note that there are several sources of uncertainty, including model differences as well as internal climate variability. Our quoted uncertainties include both. However, if we had three realizations for each model, the role of internal climate variability could be better isolated. Although three realizations are probably not enough to truly quantify this source of uncertainty, which is the goal of large ensemble projects. Similarly, additional models would also allow improved quantification of the uncertainty due to model differences. For a given variable we analyze, the uncertainty across realizations for models with multiple runs is comparable to the uncertainty across models. For example, in terms of the total global surface temperature change (Section 3), we quote a multi-model mean 95% confidence interval of 0.12 K. Models with multiple realizations yield corresponding values of 0.02, 0.08, 0.10, 0.11, 0.15, 0.22, which yields an average of 0.11 K. Similarly, in terms of the total global land $PM_{2.5}$ change, we quote a multi-model mean 95% confidence interval of 0.32 $\mu$g m$^{-3}$. Models with multiple realizations yield corresponding values of 0.08, 0.09, 0.16, 0.25, 0.29, which yields an average of 0.17 $\mu$g m$^{-3}$. Furthermore, if all models and realizations are used, our uncertainty estimates are reduced (due, in part, to more data). For example, the 95% confidence interval for total global surface temperature change decreases from the quoted 0.12 to 0.07 K. The 95% confidence interval for the total global land $PM_{2.5}$ change decreases from 0.32 to 0.24 $\mu$g m$^{-3}$.

Climate variables analyzed include monthly mean surface temperature (Ts) and precipitation (Precip). Surface temperature and precipitation are analyzed as these are arguably two of the most important climate variables. Changes in surface temperature are particularly relevant in the context of climate mitigation, as the goal of the Paris Agreement is to keep the increase in global mean surface temperature to well below 2°C above preindustrial values (IPCC, 2018). Precipitation, and fresh water resources in general, are important to both human society and ecosystems.

As discussed in the Introduction, both $PM_{2.5}$ and ozone are commonly used indicators of air quality, and both have been associated with adverse human health impacts (WHO, 2016; Cohen et al., 2017; Butt et al., 2017). Air quality is therefore quantified from surface $PM_{2.5}$ and surface $O_3$. These monthly mean fields are obtained from the model level closest to the surface. Unfortunately, few models archived sub-monthly aerosol or ozone data, so we are unable to analyze changes in daily or sub-daily maximum $PM_{2.5}$ or $O_3$ pollution. Furthermore, only four models directly archive $PM_{2.5}$ (with differing methodologies), and not all models include the same aerosol species (e.g., nitrate aerosol; Supplement). Thus, we approximate $PM_{2.5}$ in all models using the following equation (Fiore et al., 2012; Silva et al., 2017): $PM_{2.5} = BC + OA + SO_4 + 0.1\text{xDU} + 0.25\text{xSS}$, where BC is black carbon, OA is organic aerosol, $SO_4$ is sulfate aerosol, DU is dust and SS is sea salt. This formula assumes 100% of the BC, OA and $SO_4$ is fine mode, whereas 25% of the sea salt and 10% of the dust is fine mode. The SS and DU factors will be dependent on the model and its size distribution. In the case of CNRM-ESM2-1, sensitivity tests were used to

estimate a much smaller SS factor of 0.01. This smaller factor addresses the large SS size range of up to 20 $\mu$m in this model (P. Nabat 2019, personal communication, November 27th). Although this approach likely introduces some uncertainties (see Section 3.1), it provides first and foremost an estimate of $PM_{2.5}$ for all models, as well as a consistent estimate for all models.

CMIP6 model evaluation of air quality metrics, including surface $O_3$ and $PM_{2.5}$ (as approximated here), is quantified in
a companion paper (Turnock et al., 2020). To summarize, CMIP6 models generally underestimate $PM_{2.5}$ over most regions relative to ground based observations from the Global Aerosol Synthesis and Science Project (GASSP) (Reddington et al., 2017). This in part is due to the absence of nitrate aerosol, and may also be related to misrepresentation of secondary organic aerosol. A similar $PM_{2.5}$ underestimation occurs over Europe and North America relative to the Modern-Era Retrospective Analysis for Research and Applications, version 2 (MERRA2) aerosol reanalysis product (Buchard et al., 2017; Randles
et al., 2017). In contrast, CMIP6 models overestimate $PM_{2.5}$ relative to MERRA2 over south and east Asia, contrary to the evaluation using GASSP observations. Compared to surface $O_3$ measurements from Tropospheric Ozone Assessment Report (TOAR) (Schultz et al., 2017), CMIP6 models consistently overestimate surface ozone during both summer and winter across most regions, potentially due to the coarse resolution of global models simulating excess $O_3$ production.

Perhaps more important than changes in the mean of a climate variable are changes in its extremes. Heat waves, for example,
are a major cause of weather-related fatalities. Thus, we also analyze climate extremes including the hottest day (monthly maximum value of daily maximum surface temperature), wettest day (monthly maximum 1-day surface precipitation) and consecutive dry days (CDD), defined as the maximum annual number of consecutive days with surface precipitation < 1 mm day$^{-1}$. We focus on these three extreme indices since they are frequently used metrics for temperature and precipitation extremes. Prior observational analyses have shown significant increases in the hottest and wettest day, and decreases in CDD
over the latter half of the 20th century (Donat et al., 2013a, b). Climate extremes are based on daily data, and are calculated at each grid box and then spatially averaged.

## 2.3 Future Scenarios: SSP3-7.0 and SSP3-7.0-lowNTCF

As part of ScenarioMIP, a set of future emissions pathways have been developed for CMIP6 (Eyring et al., 2016). These scenarios, referred to as Shared Socio-economic Pathways (SSPs) (O'Neill et al., 2014; van Vuuren et al., 2014; Gidden et al.,
2019), link socioeconomic and technological innovation to provide future trajectories of emissions, including different levels of controls on air quality pollutants. The medium strength of pollution control corresponds to current legislation (CLE) until 2030 and progresses three-quarters of the way towards maximum technically feasible reduction (MTFR) thereafter. Strong pollution control exceeds CLE and progresses ultimately towards MTFR. Weak pollution control assumes delays to the implementation of CLE and makes less progress towards MTFR than the medium scenario (Rao et al., 2017). The rate of progress is different
for high, medium and low-income countries. By encompassing a wide range of possible futures, these scenarios provide a large sample space of potential emissions through the 21st century.

To detect the largest signal, AerChemMIP uses the SSP3-7.0 "Regional Rivalry" without climate policy ($\sim$7.0 W m$^{-2}$ at 2100) (Fujimori et al., 2017) as the reference scenario, which has the highest levels of NTCFs and "weak" levels of air quality control measures (O'Neill et al., 2014; Rao et al., 2017). The perturbation scenario SSP3-7.0-lowNTCF uses the same socio-

economic scenario, but with "strong" levels of air quality control measures (Gidden et al., 2019). Basically, the emissions drivers (population, GDP, energy and land-use) are based on SSP3, but the emissions factors of air pollutants that are related to NTCFs are associated with a Sustainability pathway represented by SSP1 in conjunction with the stringent climate policy equivalent of stabilizing the radiative forcing to around 2.6 W m$^{-2}$. Assumptions include the following: SSP3-7.0-lowNTCF can reduce $CH_4$ as if SSP1's stringent climate mitigation policy is implemented in the SSP3 world; SSP1's air pollutant legislation and technological progress can be achieved in the SSP3 world; other species (e.g., CFCs, HFCs and $SF_6$) are identical to the SSP3 baseline. Although methane reductions are included in the lowNTCF scenario, they are not included in the lowNTCF experiment. This allows quantification of the aerosol and ozone effects alone; similar SSP3-7.0-lowNTCF experiments that are analogous to those presented here, but also include the methane reductions, will also be performed and analyzed in subsequent work. Differences between these two scenarios are designed to evaluate a SSP3 world in which NTCF-related policies are enacted in the absence of other GHG-related climate policies. Moreover, our results (e.g., the magnitude of the surface temperature increase) represent an upper bound as our baseline scenario lacks climate policy and contains the highest levels of NTCFs.

Differences in climate, effective radiative forcing, chemical composition and air quality between the two scenarios will be solely due to the alternative air quality control measures. These experiments cover the time frame from 2015 to 2055, as this is when reductions in aerosol and ozone precursor emissions are expected to be significant, particularly in some world regions. Here, we define NTCF mitigation as the difference between the strong (low NTCF) and weak (high NTCF) air quality control scenarios (i.e., SSP3-7.0-lowNTCF minus SSP3-7.0). Although methane reductions are included in the strong air quality control scenario, AerChemMIP protocol specifies unchanged levels of WMGHGs, including methane, between the strong and the weak air quality control simulations (Collins et al., 2017). Thus, our results quantify non-methane NTCF mitigation (aerosols and ozone only).

Figure 1 shows the 2015-2055 global mean time series of $CO_2$, aerosol species and gaseous precursor emissions for SSP3-7.0 (weak air quality control) and SSP3-7.0-lowNTCF (strong air quality control). Emissions shown here comes directly from the CMIP6 forcing datasets, which were downloaded from the input datasets for Model Intercomparison Project (input4MIPS) served by the Earth System Grid Federation. We note that the emissions data is decadal after 2015, with monthly values for the year 2015, 2020, 2030, 2040, 2050, 2060, etc. We estimate the emissions in 2055 as the mean of the emissions in 2050 and 2060 at each grid box. Only weak air quality control $CO_2$ and $CH_4$ emissions are shown, as AerChemMIP simulations include the same change in $CO_2$ and $CH_4$ emissions based on the weak air quality control scenario. By 2055, $CO_2$ and $CH_4$ increase by 65% and 50% (relative to 2015), respectively. In contrast to $CO_2$ and $CH_4$, however, very different non-methane NTCF evolution occurs. Under weak air quality control, global emissions of all aerosols and gaseous precursors (except $SO_2$) increase by 5-15% by 2055. In contrast, strong air quality control yields strong emission reductions in all species, ranging from ~30% for VOCs to 55% for $SO_2$. Thus, NTCF mitigation (SSP3-7.0-lowNTCF−SSP3-7.0) yields emission reductions of all aerosols and gaseous precursors by ~40-55%.

The corresponding 2015-2055 regional emission trends (relative to 2015) are shown in Figure 2. As with climate and air quality trends, emission trends are estimated using least-squares regression. Consistent with the global mean time series of

emissions (Fig. 1), $CO_2$ emissions increase under weak air quality control (and in both sets of AerChemMIP simulations), with larger increases in south and north Africa, south Asia, and southeast Asia. Similarly, $CH_4$ emissions increase in all world regions under weak air quality control (and in both sets of AerChemMIP simulations), with larger increases in south and north Africa, and south Asia. Most world regions also show increases in BC, $SO_2$ and organic carbon (OC) under weak air quality control, but strong decreases under strong air quality control. NTCF mitigation (strong minus weak air quality control) shows large ($\sim$20% decade$^{-1}$) BC decreases in central America, central and north Asia, east Asia and southeast Asia. Most world regions exhibit a 10-20% decade$^{-1}$ reduction in $SO_2$ emissions under NTCF mitigation, with a large decrease in south Asia at $-28\%$ decade$^{-1}$. Similarly, OC and CO emissions decrease by $\sim$10-20% decade$^{-1}$. Relatively large OC reductions also occur in east Asia, south Asia and southeast Asia. $NO_x$ and VOC emissions also decrease in all world regions under NTCF mitigation (although this is only a decrease relative to non-mitigated emissions for $NO_x$ in south Asia and for VOC in east Asia).

## 3   Results

### 3.1   Global Climate and Air Quality Trends

Figure 3 shows the 2015-2055 global annual mean time series for air quality under NTCF mitigation. By design, NTCF mitigation leads to significant decreases in air pollution, in terms of both surface $PM_{2.5}$ and $O_3$. All models yield significant global mean decreases in both quantities, with an overall MMM decrease of $-0.23$ $\mu$g m$^{-3}$ decade$^{-1}$ for $PM_{2.5}$ and $-1.19$ ppb decade$^{-1}$ for $O_3$ (Table 1). Over the 2015-2055 year time period, these rates of change correspond to global mean decreases of $-0.92$ $\mu$g m$^{-3}$ and $-4.76$ ppb, respectively. Larger $PM_{2.5}$ decreases occur over land only at $-2.20$ $\mu$g m$^{-3}$, whereas similar $O_3$ decreases occur over land only at $-4.55$ ppb. Similar $PM_{2.5}$ trends occur in Aer+O3 and Aer models over land only ($-0.59$ versus $-0.44$ $\mu$g m$^{-3}$ decade$^{-1}$, respectively), as well as over both land and ocean ($-0.26$ versus $-0.16$ $\mu$g m$^{-3}$ decade$^{-1}$, respectively). Note that the MMM over all models for $O_3$ does not include Aer models, as they yield negligible change in surface ozone (by design).

As mentioned in the Methods section, as only four models directly archive $PM_{2.5}$ (with differing methodologies), and not all models include the same aerosol species (Supplement), we approximate $PM_{2.5}$. Comparing estimated $PM_{2.5}$ trends to those from the actual $PM_{2.5}$ as calculated and archived by four models (GFDL-ESM4, NorESM2-LM, MRI-ESM2-0 and MPI-ESM1-2-HAM) yields reasonably good results. The global annual multi-model mean trend in estimated (actual) $PM_{2.5}$ for this four model subset is $-0.24$ ($-0.28$) $\mu$g m$^{-3}$ decade$^{-1}$ (Supplementary Figure 1). Over land only, the corresponding trends are $-0.56$ ($-0.65$) $\mu$g m$^{-3}$ decade$^{-1}$. Thus, the estimated global mean and land-only $PM_{2.5}$ trends are about 85% as large as those based on archived $PM_{2.5}$ (underestimation by a similar amount exists in all four models, with the largest underestimation in GFDL-ESM4). Larger differences exist in some world regions, particularly south Asia, where the estimated (actual) $PM_{2.5}$ is $-4.08\pm0.70$ ($-4.71\pm1.36$) $\mu$g m$^{-3}$ decade$^{-1}$ (Supplementary Figure 2). Some of this underestimation is due to the aforementioned lack of nitrate and ammonium aerosol in our estimated $PM_{2.5}$. However, other factors also contribute, as

the estimated $PM_{2.5}$ trends in all four models underestimate the actual $PM_{2.5}$ trends, but not all of these models include nitrate and ammonium species.

GFDL-ESM4 is the lone model that archived nitrate aerosol data. Globally (over land only), nitrate decreases by $-0.04$ ($-0.12$) $\mu$g m$^{-3}$ decade$^{-1}$, with maximum decrease over east Asia and in particular south Asia (Supplementary Figure 3). These trends are 17 and 20% (13 and 15%) of the magnitude of the estimated (actual) global and land-only $PM_{2.5}$ trend. GFDL-ESM4 also archives ammonium, and similar changes occur (Supplementary Figure 3). Globally (over land only), ammonium decreases by $-0.05$ ($-0.12$) $\mu$g m$^{-3}$ decade$^{-1}$, with maximum decreases over both south Asia and east Asia. These trends are 21 and 20% (16 and 15%) of the magnitude of the estimated (actual) global and land-only $PM_{2.5}$ trend. Thus, excluding nitrate and ammonium in GFDL-ESM4 leads to $\sim$30-40% underestimation of the global and land-only PM2.5 trend. The relatively large decreases in nitrate and ammonium in south Asia helps to explain the relatively large difference in estimated and actual $PM_{2.5}$ trend in this region (Supplementary Figure 2). In addition to GFDL-ESM4, CESM2-WACCM also archives ammonium (Supplementary Figure 3). Here, however, the global and land-only ammonium trends are an order of magnitude smaller than those in GFDL-ESM4, which leads to $\sim$1% underestimation of the corresponding (estimated) PM2.5 trends.

The 2015-2055 global annual mean time series for climate variables under NTCF mitigation are shown in Figure 4. All but one model (MIROC6) shows significant global annual mean surface warming in response to NTCF mitigation (Table 1 lists the trends for each model). Averaged over all models, global mean surface warming is 0.06 K decade$^{-1}$, or 0.25 K over the 2015-2055 time period (Table 1). We note that this warming will continue past 2055, as these transient simulations have not reached radiative equilibrium. Similar conclusions exist over land only, where the multi-model mean (MMM) warming is even larger at 0.36 K over the entire time period (Table 1). Enhanced land warming is consistent with the land-sea warming contrast (Sutton et al., 2007; Joshi et al., 2008), which may also act to increase aerosol burden itself (Allen et al., 2016, 2019b), implying a climate change penalty to air quality. Interestingly, models that include both aerosol and ozone reductions (Aer+O3) yield similar surface warming relative to the models that include aerosol reductions (Aer) alone (0.07 versus 0.06 K decade$^{-1}$, respectively). Although this could be due to several factors (e.g., small sample size, internal climate variability, different model parameterizations, feedbacks, etc.) it suggests weak surface cooling due to reductions in ozone. Such an interpretation is consistent with the negative forcing from aerosol increases dominating the positive forcing due to ozone increases over the historical period (Naik et al., 2013). Simulations with a single model, running both coupled and uncoupled chemistry experiments, would help isolate this effect.

Warming in response to NTCF mitigation is consistent with the corresponding increase in ERF. All but two models (BCC-ESM1, GFDL-ESM4) yield a significant increase in ERF, with a MMM of 0.44 W m$^{-2}$ over the entire time period (Table 1). Over land only, this increases to 0.59 W m$^{-2}$. Although not significant, Aer+O3 models yield a weaker trend in global mean ERF than Aer models, at 0.07 versus 0.17 W m$^{-2}$ decade$^{-1}$. This is consistent with ozone reductions driving a decrease in ERF in Aer+O3 models, offsetting part of the ERF increase due to aerosol reductions (Turnock et al., 2019).

All models also yield a significant increase in global annual mean precipitation (Table 1), with an overall MMM of 0.008 mm day$^{-1}$ decade$^{-1}$. Aer+O3 and Aer models yield similar increases in global mean precipitation at 0.009 and 0.005 mm

day$^{-1}$ decade$^{-1}$, respectively. Somewhat less robust results occur over land only. Although all models yield an increase in precipitation over land, it is only significant in four models.

Similar, but less robust responses also occur in climate extremes, particularly those based on precipitation. Globally significant increases in the surface temperature of the hottest day occur in all but one model (MIROC6 is the exception). The multi-model mean also yields a significant trend at 0.06 K decade$^{-1}$. The wettest day significantly increases in about half of the models and in the overall MMM at 0.053 mm day$^{-1}$ decade$^{-1}$. A mixed signal exists for CDD, with four models yielding a positive trend and three models yielding a negative trend. The overall MMM yields 0.08 days per year decade$^{-1}$, but lacks significance.

Thus, from a global mean perspective, NTCF mitigation leads to significant improvements in air quality based on both PM$_{2.5}$ and O$_3$, but also significant climate change in most metrics. This includes increases in surface temperature and precipitation, as well as corresponding increases in most climate extremes, particularly the hottest day and to lesser extent the wettest day. Except for surface temperature and the hottest day, less robust results generally occur over land only. CDD yields a mixed signal, with lack of significance in the multi-model mean.

### 3.2 Regional Climate and Air Quality Trends

Figure 5 shows the regional climate and air pollution trends for weak and strong air quality control and the effect of NTCF mitigation. We include both Aer and Aer+O3 models in this analysis to maximize the signal to noise ratio (except for ozone changes). The aforementioned response differences between these two model subsets are generally not significant. Consistent with increased aerosol and precursor gas emissions (Figures 1-2), air quality metrics generally show significant increases under weak air quality control, particularly O$_3$ where all 12 world regions exhibit an increase. In contrast, strong air quality control yields decreases in both PM$_{2.5}$ and O$_3$ for nearly all world regions. The overall effect of NTCF mitigation is thus a robust decrease in air pollution, in terms of both PM$_{2.5}$ and O$_3$, over all 12 world regions, as well as the Arctic, Northern Hemisphere (NH) midlatitudes, and Tropics. Over all land surfaces, the PM$_{2.5}$ decrease is $-0.55\pm0.08$ $\mu$gm$^{-3}$ decade$^{-1}$. Regionally, decreases in PM$_{2.5}$ range from $-0.05\pm0.01$ $\mu$gm$^{-3}$ decade$^{-1}$ over Canada to $-3.8\pm0.69$ $\mu$gm$^{-3}$ decade$^{-1}$ in south Asia. Relatively large PM$_{2.5}$ decreases also occur over east Asia, southeast Asia and north Africa at $-2.1\pm0.27$ $\mu$gm$^{-3}$ decade$^{-1}$, $-0.78\pm0.16$ $\mu$gm$^{-3}$ decade$^{-1}$, and $-0.82\pm0.20$ $\mu$gm$^{-3}$ decade$^{-1}$, respectively. The relatively large PM$_{2.5}$ decreases over east Asia, southeast Asia, and south Asia are generally consistent with the relatively large reductions in aerosol species, including BC, SO$_4$ and OC (Figure 2).

Similar results exist for O$_3$, with a robust decrease over land of $-1.11\pm0.22$ ppb decade$^{-1}$. Regionally, O$_3$ decreases range from $-2.41\pm0.33$ ppb decade$^{-1}$ over central America and $-1.97\pm0.20$ ppb decade$^{-1}$ over southeast Asia to $-0.86\pm0.11$ ppb decade$^{-1}$ over Australia. Relatively large O$_3$ decreases also occur over south Asia ($-1.55\pm0.93$ ppb decade$^{-1}$), as well as north Africa ($-1.7\pm0.25$ ppb decade$^{-1}$). Notably, a weak O$_3$ decrease occurs in east Asia ($-0.45\pm0.51$ ppb decade$^{-1}$), which may be related to relatively weak VOC reductions (Figure 2). In addition to significant reduction in the Arctic, the other latitudinal bands also exhibit significant reductions in O$_3$.

Over all 12 world regions, significant surface warming occurs in both the weak and strong air quality control scenarios, due to continued increases in $CO_2$ (and $CH_4$). More importantly, NTCF mitigation−due to reduced cooling from reductions in non-absorbing aerosol (e.g., sulfate)−also yields significant warming, with a significant increase in land-only surface temperature of $0.09\pm0.02$ K decade$^{-1}$. Significant warming also occurs in all but one world region (Australia is the lone exception) due to NTCF mitigation, ranging from $0.05\pm0.02$ K decade$^{-1}$ over southeast Asia to $0.16\pm0.05$ K decade$^{-1}$ over central and north Asia. Relatively large warming also occurs over east Asia ($0.11\pm0.05$ K decade$^{-1}$) and south Asia ($0.12\pm0.04$ K decade$^{-1}$; see also Supplementary Figure 4). Furthermore, large warming of the Arctic (60-90N) occurs ($0.15\pm0.09$ K decade$^{-1}$), particularly in the East Siberian and Beaufort Seas, north of western Canada/Alaska and around the Canadian Arctic Archipelago (Figure 6). This result is consistent with recent studies showing high Arctic sensitivity to aerosol reductions (Acosta Navarro et al., 2016; Lewinschal et al., 2019; Westervelt et al., 2020). Other latitudinal bands also significantly warm, including the NH midlatitudes (30-60N), Tropics (30S-30N), and Southern Hemisphere (SH) midlatitudes (60S-30S) at $0.10\pm0.03$, $0.05\pm0.01$, $0.03\pm0.02$ K decade$^{-1}$, respectively.

Warming is consistent with the increase in ERF, with most world regions yielding significant positive ERF trends. Little correspondence exists between regions that warm the most and their ERF trend. This is not necessarily surprising, as forcing and response do not need to occur in the same regions, due to climate feedbacks, remote teleconnection and other processes. For example, central and north Asia and the Arctic warm the most, but there is not a particularly large increase in their regional ERF. Similarly, southeast Asia warms the least, but this region features a relatively large ERF increase.

Significant increases in the hottest day also occur, with larger increases under strong relative to weak air quality controls. NTCF mitigation yields significant increases in the hottest day for all but two world regions (Australia and south America are the exceptions). A significant increase in the hottest day also occurs over all land regions ($0.09\pm0.03$ K decade$^{-1}$), with five of six models yielding a significant increase (Table 1; MIROC6 is the exception). Thus, NTCF mitigation unmasks the warming due to GHG increases resulting in robust increases in both surface air temperature and the hottest day over nearly all world regions. We note that the lone area with cooling is the north Atlantic (around Iceland and southwest of Svalbard; Fig. 6), which may be associated with a weakening of the Atlantic Meridional Overturning Circulation (AMOC) (Delworth and Dixon, 2006; Cai et al., 2006; Menary et al., 2013). Figures 6d-f show that this cooling is a robust feature, with ∼70% of the models yielding cooling here.

Over all land surfaces, a significant precipitation increase also occurs in both scenarios at $0.012\pm0.005$ and $0.022\pm0.006$ mm day$^{-1}$ decade$^{-1}$ under weak and strong air quality control, respectively. Thus, NTCF mitigation−by unmasking GHG-induced warming−also yields a significant increase in land precipitation at $0.011\pm0.003$ mm day$^{-1}$ decade$^{-1}$. The effect of NTCF mitigation on precipitation over individual world regions, however, has mixed significance and ranges from $0.003\pm0.035$ mm day$^{-1}$ decade$^{-1}$ over Australia to $0.044\pm0.022$ mm day$^{-1}$ decade$^{-1}$ over south Asia. Note that some world regions exhibit decreases in precipitation under both weak and strong air quality control (e.g., central America), such that NTCF mitigation yields a weaker decrease (as opposed to an absolute increase). In addition to south Asia, a significant precipitation increase also occurs over central and north Asia ($0.008\pm0.005$ mm day$^{-1}$ decade$^{-1}$), east Asia ($0.038\pm0.014$ mm day$^{-1}$ decade$^{-1}$) and the Arctic ($0.010\pm0.005$ mm day$^{-1}$ decade$^{-1}$). Although southeast Asia also exhibits a relatively large increase in precipitation,

it is not significant ($0.019\pm0.041$ mm day$^{-1}$ decade$^{-1}$). Both south and north Africa yield precipitation increases, but the bulk of the African precipitation increase occurs over East Africa (Supplementary Figure 5). From a latitudinal perspective, in addition to the Arctic, the NH midlatitudes, Tropics and SH midlatitudes all experience a significant increase in precipitation at $0.010\pm0.005$, $0.011\pm0.001$, and $0.004\pm0.001$ mm day$^{-1}$ decade$^{-1}$, respectively. Thus, NTCF mitigation generally increases precipitation in most world regions (although, in some regions, this is a smaller decrease) but the signal is less robust than that for surface temperature. Furthermore, in agreement with prior studies (Levy et al., 2013; Westervelt et al., 2017; Zhao et al., 2018; Westervelt et al., 2018; Scannell et al., 2019), precipitation increases in several monsoon regions, including east Africa, south Asia, and east Asia.

Precipitation extremes, including the wettest day and in particular CDD, also exhibit regional uncertainty under NTCF mitigation, with most regions lacking a robust response. Similar to the significant increases in mean precipitation, significant increases in the wettest day also occur in central and north Asia, east Asia, south Asia, and the Arctic. The NH midlatitudes and Tropics (but not the SH midlatitudes) also experience a robust increase in the wettest day. NTCF mitigation also yields robust CDD increases in south America and south Africa, and robust CDD decreases in Canada and the Arctic. Outside of the Arctic, no other latitudinal bands yield a robust CDD response under NTCF mitigation.

### 3.3 Seasonal Climate and Air Quality Trends

Figure 7 shows the regional surface temperature, precipitation and air quality responses during June-July-August (JJA) and December-January-February (DJF). Seasonal air pollution, including both $O_3$ and $PM_{2.5}$, exhibit robust decreases in nearly all world regions under NTCF mitigation. Over land regions, slightly larger $O_3$ decreases occur during JJA relative to DJF, at $-1.41\pm0.16$ ppb decade$^{-1}$ and $-0.86\pm0.34$ ppb decade$^{-1}$, respectively. This seasonal contrast is more pronounced over the NH midlatitudes, where the JJA (DJF) decrease is $-1.87\pm0.17$ ($-0.72\pm0.52$) ppb decade$^{-1}$. In contrast, slightly larger $PM_{2.5}$ decreases occur during DJF relative to JJA, at $-0.67\pm0.12$ $\mu$g m$^{-3}$ decade$^{-1}$ and $-0.48\pm0.08$ $\mu$g m$^{-3}$ decade$^{-1}$, respectively. As with the annual mean, the largest JJA and DJF $O_3$ reductions occur over central America and southeast Asia (and north Africa during DJF). The lone regional increase in $O_3$ occurs during DJF in east Asia at $0.95\pm0.52$ ppb decade$^{-1}$. The largest JJA decreases in $PM_{2.5}$ occur in east Asia ($-1.64\pm0.31$ $\mu$g m$^{-3}$ decade$^{-1}$) and south Asia ($-1.67\pm0.32$ $\mu$g m$^{-3}$ decade$^{-1}$) . These regions also exhibit large DJF decreases in $PM_{2.5}$, particularly south Asia at $-5.55\pm1.2$ $\mu$g m$^{-3}$ decade$^{-1}$.

NTCF mitigation yields similar warming in both seasons (see also Supplementary Figures 6-7). Over all land surfaces, JJA warming is $0.09\pm0.02$ K decade$^{-1}$; DJF warming is $0.09\pm0.04$ K decade$^{-1}$. Consistent with the annual mean warming, relatively large JJA warming also occurs in central and north Asia ($0.16\pm0.06$ K decade$^{-1}$), south Asia ($0.10\pm0.05$ K decade$^{-1}$) and east Asia ($0.10\pm0.06$ K decade$^{-1}$), as well as Canada ($0.14\pm0.05$ K decade$^{-1}$). DJF warming is largest in similar regions as JJA, including central and north Asia and south Asia ($0.20\pm0.12$ and $0.13\pm0.04$ K decade$^{-1}$) and east Asia ($0.13\pm0.12$ K decade$^{-1}$). Arctic warming is most pronounced during DJF, where the rate of warming is about double that during JJA ($0.23\pm0.16$ versus $0.12\pm0.05$ K decade$^{-1}$). Similar JJA and DJF warming occurs for the NH midlatitudes ($0.11$ versus $0.10\pm0.03$ K decade$^{-1}$), Tropics ($0.05$ versus $0.04\pm0.02$ K decade$^{-1}$) and SH midlatitudes ($0.03\pm0.02$ K decade$^{-1}$ for

both seasons). As with the annual mean warming, central and north Asia, east Asia, and south Asia generally warm the most during JJA and DJF, with large Arctic warming during DJF.

Regional seasonal precipitation responses continue to exhibit relatively large uncertainty, as most world regions lack a robust response (see also Supplementary Figures 8-9). Central and north Asia, east Asia and south Asia yield robust JJA increases in precipitation under NTCF mitigation at $0.015\pm0.008$, $0.053\pm0.034$, and $0.089\pm0.047$ mm day$^{-1}$ decade$^{-1}$, respectively. The increase in south and east Asia precipitation is consistent with aerosol reductions driving enhanced monsoonal flow. Interestingly, there is also a significant increase in south Asian precipitation during DJF. Canada and north Africa in particular also exhibit robust increases in DJF precipitation. As with the annual mean, most of the increase in DJF precipitation over Africa occurs in east Africa (Supplementary Figure 9).

Similar results generally exist for the other seasons, March-April-May (MAM) and September-October-November (SON) (Supplementary Figure 10). The largest decrease in $O_3$ occurs in central America, south Asia and southeast Asia, as well as north Africa. The largest $PM_{2.5}$ decreases occur in east Asia, south Asia, and southeast Asia. Over all land surfaces, MAM and SON surface warming are both $0.09\pm0.02$ K decade$^{-1}$. Maximum MAM (SON) regional warming occurs in central and north Asia (Arctic) at $0.18\pm0.05$ ($0.17\pm0.09$) K decade$^{-1}$. Relatively large MAM warming also occurs in east Asia ($0.13\pm0.07$ K decade$^{-1}$) and south Asia ($0.11\pm0.06$ K decade$^{-1}$); relatively large SON warming occurs in Canada ($0.15\pm0.06$ K decade$^{-1}$) and Europe ($0.13\pm0.05$ K decade$^{-1}$). Precipitation responses are again less robust, although east Asia experiences robust increases in both seasons ($0.03\pm0.02$ in MAM and $0.06\pm0.03$ mm day$^{-1}$ decade$^{-1}$ in SON). Relatively large SON precipitation increases also occur for central America and south Asia.

## 4  Conclusions

Under the experimental protocols of ScenarioMIP (O'Neill et al., 2016) and AerChemMIP (Collins et al., 2017), we have analyzed future chemistry-climate simulations to assess the impact of non-methane NTCF mitigation of climate and air quality from 2015-2055. Simulations show robust decreases in air pollution in nearly all world regions. Over global land, surface $PM_{2.5}$ and $O_3$ decrease by $-2.2$ $\mu$g m$^{-3}$ and $-4.6$ ppb, respectively, with larger reductions in some world regions including south and southeast Asia. However, NTCF mitigation unmasks the warming due to GHG increases, resulting in additional global warming and precipitation increases of 0.25K and 0.03 mm day$^{-1}$, respectively. Similarly, increases in extreme weather indices also occur, including the hottest and wettest day. All but one world region (minus Australia) yields robust warming in response to NTCF mitigation, with the largest warming (and wetting) occurring over Asia, including central and north Asia, east Asia and south Asia. Relatively large warming also occurs over the Arctic at 0.59K, more than double the global mean warming. Interestingly, models that include both aerosol and ozone reductions (Aer+O3) yield similar warming (and wetting) relative to models that include aerosol reductions alone (Aer). This suggests a weak cooling effect due to ozone reductions, or other possible effects from interactive chemistry and aerosol that need to be further explored. For example, aerosol formation may be reduced due to changes in oxidants (from $O_3$ reductions), which would lead to more surface warming in Aer+O3. Simulations with a single model, running both coupled and uncoupled chemistry experiments, would help isolate this effect.

We also reiterate that few models include nitrate aerosol, which implies an underestimation bias in the climate responses shown here.

Our results are consistent with several studies that have shown aerosol reductions will unmask GHG warming, resulting in large, near-term increases in global surface temperature and precipitation (Brasseur and Roeckner, 2005; Andreae et al., 2005; Ramanathan and Feng, 2008; Raes and Seinfeld, 2009; Kloster et al., 2010; Arneth et al., 2009; Matthews and Zickfeld, 2012; Rotstayn et al., 2013; Wu et al., 2013; Westervelt et al., 2015; Salzmann, 2016; Hienola et al., 2018; Richardson et al., 2018; Samset et al., 2018; Lelieveld et al., 2019). Shindell and Smith (2019), however, show that the time required to transform power generation, industry and transportation leads to largely offsetting climate impacts of $CO_2$ and sulfur dioxide, implying no conflict between climate and air-quality objectives. There, a 1.5°C mitigation pathway is used, with gradual phasing out of fossil fuel combustion, which leads to relatively small change in the near-future warming. Furthermore, Shindell and Smith (2019) include methane mitigation, which compensates the relatively small near-term future warming from $SO_2$ reductions.

Our simulations, however, do not account for $CO_2$ or $CH_4$ reductions, implying the importance of simultaneous reductions in both WMGHGs and NTCFs. We note that it is difficult to reduce only the NTCF emissions while keeping $CO_2$ emissions fixed (since there are co-emitted species, including $SO_2$). If WMGHG emissions are simultaneously reduced along with non-methane NTCFs, then the increase in global surface temperature and precipitation found here will be muted (and perhaps, offset). Moreover, our results (e.g., the magnitude of the surface temperature increase) represent an upper bound as our baseline scenario lacks climate policy and contains the highest levels of NTCFs. The lowNTCF scenario, however, can be used to provide forcing and response sensitivities under current climate, which could be used by intermediate complexity models for testing out more scenarios which include complex NTCF-$CO_2$ reduction scenarios. Furthermore, the AerChemMIP SSP3-7.0-lowNTCF simulations used in this study do not account for reductions of methane, which is another NTCF, the reduction of which would promote net cooling (i.e., reduced warming). As the strong air quality control pathway includes reductions of methane, additional AerChemMIP simulations are being conducted that include the effects of all NTCFs, including aerosols, ozone precursor gases and methane. The inclusion of methane reductions will offset some of the warming reported here, and also impact tropospheric $O_3$ and air quality. Although not addressed in this study, we also note the potential role of hydrofluorocarbon (HFC) mitigation through the Kigali Amendment, particularly for the late 21st century. Efficient implementation of the Kigali Amendment and national regulations is estimated to lead to relatively small cooling (<0.07°C) by 2050, but this increases to cooling of 0.2-0.4°C by 2100 (WMO, 2018). Nonetheless, cleaning the air while keeping global warming below the 1.5-2°C Paris Agreement climate target will require simultaneous cuts in both NTCFs and carbon dioxide.

*Data availability.* CMIP6 model data can be freely downloaded from the Earth System Grid Federation (ESGF) server at https://esgf-node. llnl.gov/projects/cmip6/. The emissions data used here can also be downloaded from the ESGF via the input datasets for Model Intercomparison Projects (input4MIPs) at https://esgf-node.llnl.gov/search/input4mips/.

*Author contributions.* R.J.A. performed the analysis and wrote the paper. D.N., U.L. and I.T. performed MPI-ESM1-2-HAM simulations. P.N. and M.M. performed CNRM-ESM2-1 simulations. T.W. and J.Z. performed BCC-ESM1 simulations. N.O. and M.D. performed MRI-ESM2-0 simulations. T.T. performed MIROC6 simulations. L.E. and J-F.L. performed CESM2 and CESM2-WACCM simulations. L.H., V.N. L.S., and J. J. performed GFDL-ESM4 simulations. W.J.C., J-F.L. and M.S. originally conceived the AerChemMIP project, including the low NTCF simulations analyzed here. All authors contributed to editing the manuscript.

*Competing interests.* The authors declare that they have no competing financial interests.

*Acknowledgements.* T. Takemura was supported by the supercomputer system of the National Institute for Environmental Studies, Japan, and JSPS KAKENHI Grant Number JP19H05669. D. Neubauer acknowledges funding from the European Union's Horizon 2020 research and innovation programme project FORCeS under grant agreement No 821205. D. Neubauer and I. Tegen acknowledge a grant for computing resources from the Deutsches Klimarechenzentrum (DKRZ) under project ID 1051. The CESM project is supported primarily by the National Science Foundation. This material is based upon work supported by the National Center for Atmospheric Research, which is a major facility sponsored by the NSF under Cooperative Agreement No. 1852977. Computing and data storage resources, including the Cheyenne supercomputer (doi:10.5065/D6RX99HX), were provided by the Computational and Information Systems Laboratory (CISL) at NCAR. S. Shim was supported by the Korea Meteorological Administration Research and Development Program "Development and Assessment of IPCC AR6 Climate Change Scenario", grant agreement number 1365003000. D.O. performed NorESM2-LM simulations. M. Deushi and N. Oshima were supported by the Japan Society for the Promotion of Science (grant numbers: JP18H03363, JP18H05292, and JP20K04070) and the Environment Research and Technology Development Fund (2-1703, 2-2003, and 5-2001) of the Environmental Restoration and Conservation Agency, Japan. Finally, we thank two anonymous reviewers and the Editor for helpful comments on the initial submissions of this manuscript.

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

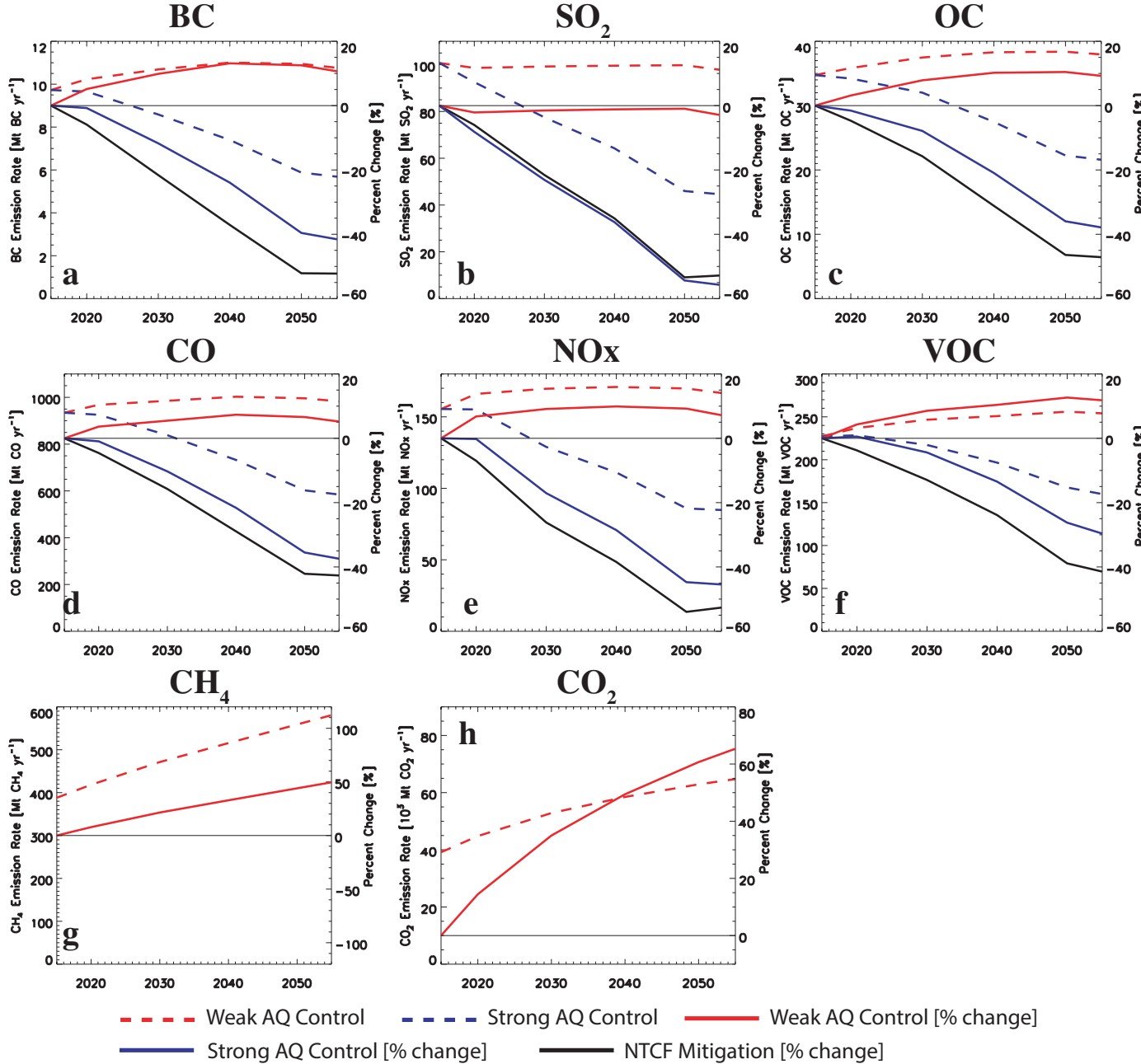

**Figure 1. 2015-2055 global mean CO$_2$, NTCF and precursor gas emissions.** Panels show (a) black carbon (BC); (b) sulfur dioxide (SO$_2$); (c) organic carbon (OC); (d) carbon monoxide (CO); (e) nitrogen oxides (NO$_x$); (f) volatile organic compounds (VOC); (g) methane (CH$_4$); and (h) carbon dioxide (CO$_2$) emissions for weak (red) and strong (blue) air quality control. Also included is the percent change (relative to 2015) for weak (red solid) and strong (blue solid) air quality control, and NTCF mitigation (black solid). Emission units for species X are Mt X yr$^{-1}$. Percent change units are %. Only weak air quality control CO$_2$ and CH$_4$ emissions are shown, as AerChemMIP simulations include the same change in CO$_2$ and CH$_4$ emissions based on the weak air quality control scenario. Emissions data comes directly from the CMIP6 forcing datasets, which were downloaded from the input datasets for Model Intercomparison Project (input4MIPS).

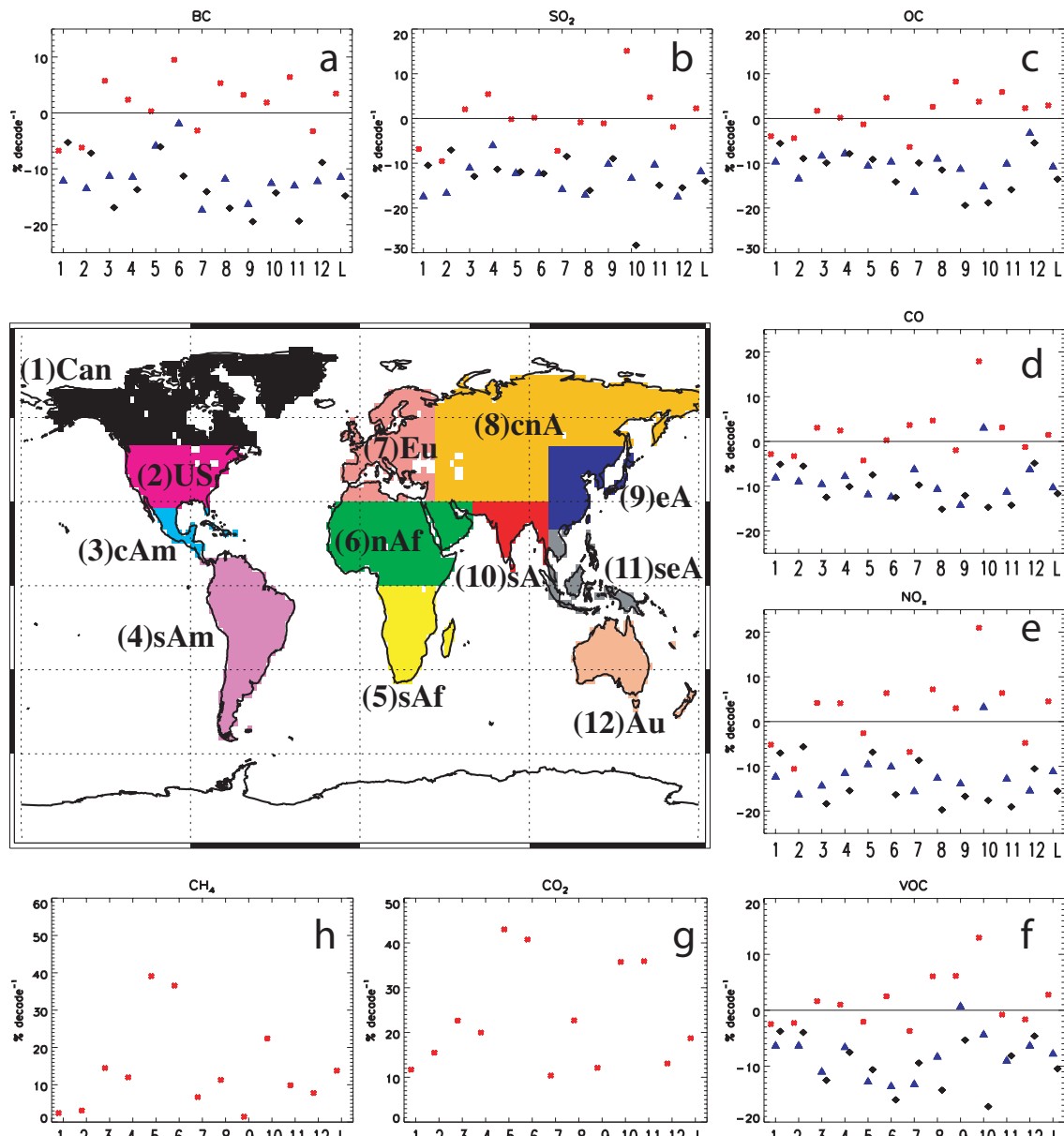

**Figure 2. 2015-2055 regional mean CO₂, NTCF and precursor gas emission trends.** Regional 2015-2055 emission trends for (a) black carbon (BC); (b) sulfur dioxide (SO₂); (c) organic carbon (OC); (d) carbon monoxide (CO); (e) nitrogen oxides (NO$_x$); (f) volatile organic compounds (VOC); (g) methane (CH₄); and (h) carbon dioxide (CO₂) for weak (red asterisks) and strong air quality control (SSP3-7.0-lowNTCF; blue triangles) and NTCF mitigation (SSP3-7.0-lowNTCF−SSP3-7.0; black diamonds). Center map shows the corresponding color coded world regions, based on Seneviratne et al. (2012). The following abbreviations are used: Canada = 1 (Can; black), United States = 2 (US; magenta), central America = 3 (cAm; sky blue), south America = 4 (sAm; purple), south Africa = 5 (sAf; yellow), north Africa = 6 (nAf; green), Europe = 7 (Eu; pink), central and north Asia = 8 (cnA; orange), east Asia = 9 (eA; navy), south Asia = 10 (sA; red), southeast Asia = 11 (seA; gray), and Australia = 12 (Au; beige). The average over these 12 land regions is abbreviated as "L". Trend units are % decade⁻¹ (relative to 2015). Only weak air quality control CO₂ and CH₄ emission trends are shown, as AerChemMIP simulations include the same change in CO₂ and CH₄ emissions based on the weak air quality control scenario. Emissions data comes directly from the CMIP6 forcing datasets, which were downloaded from the input datasets for Model Intercomparison Project (input4MIPS).

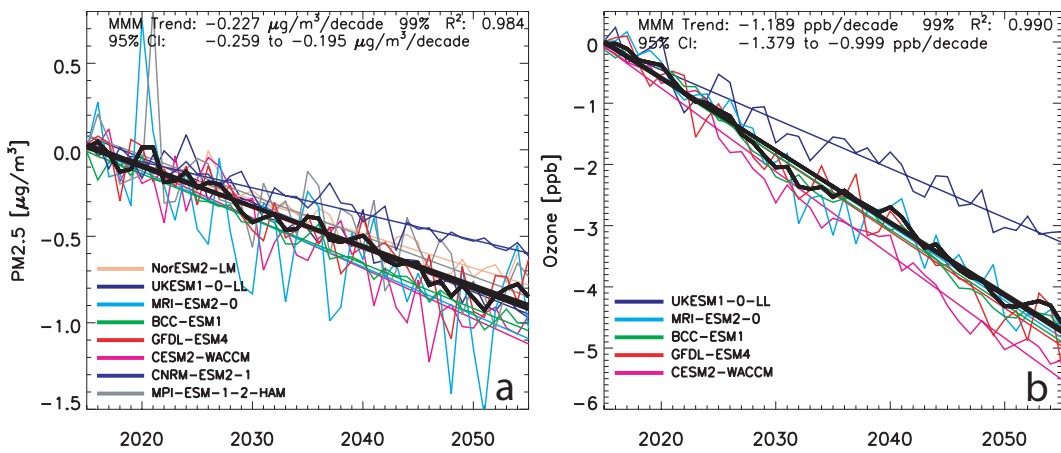

**Figure 3. 2015-2055 time series of global annual mean air pollution due to NTCF mitigation.** Panels show (a) surface particulate matter (PM$_{2.5}$) [$\mu$g m$^{-3}$] and (b) surface ozone [ppb] for NTCF mitigation. The multi-model mean time series, and the corresponding trend estimated using a weighted least-squares regression, are included as thick black lines. The multi-model mean (MMM) trend, its significance and R$^2$ value, are also included, as is the 95% confidence interval (CI). Individual model mean trends are also included as defined by the legend.

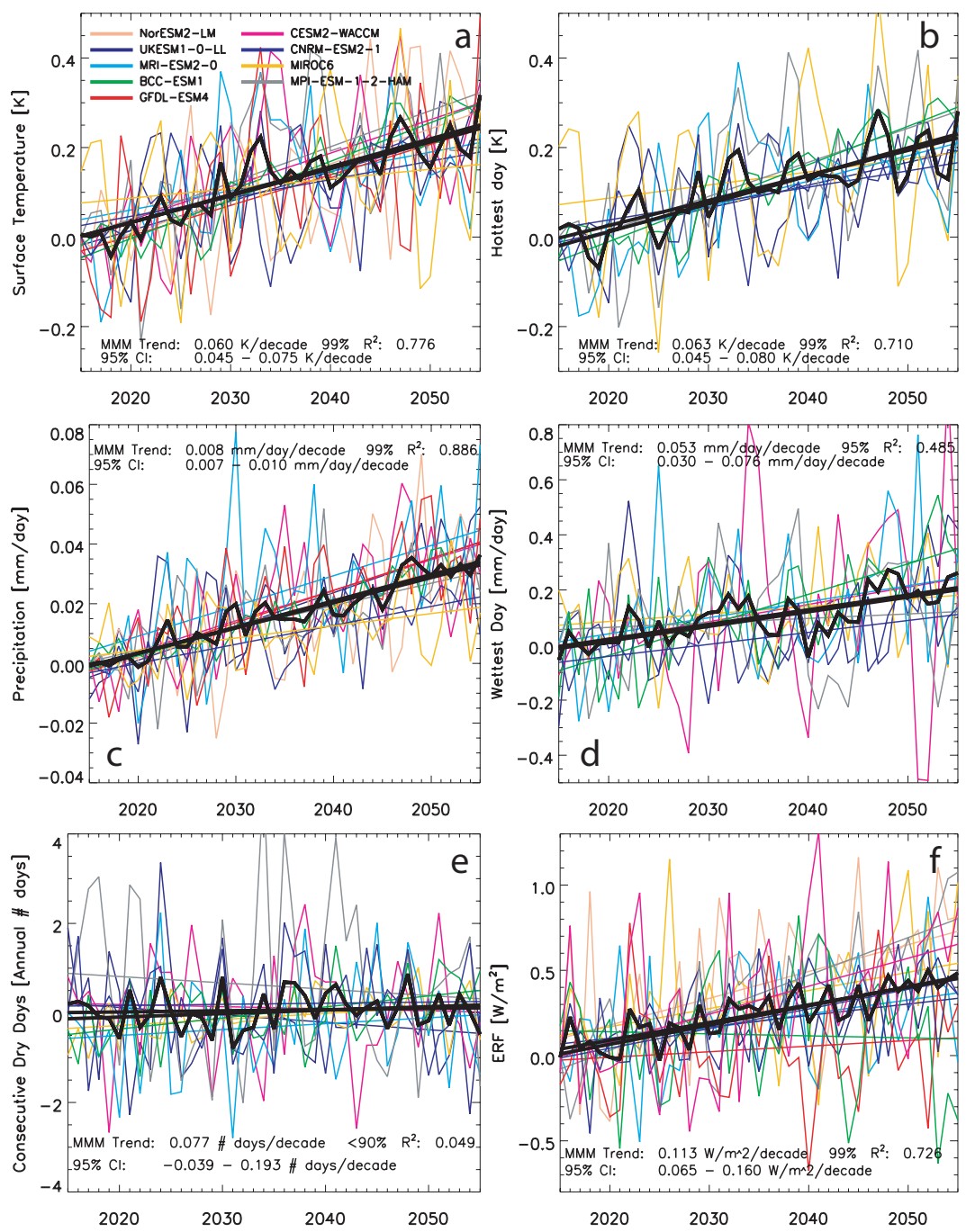

**Figure 4. 2015-2055 time series of global annual mean climate variables due to NTCF mitigation.** Panels show (a) surface temperature [K]; (b) hottest day [K]; (c) precipitation [mm day$^{-1}$]; (d) wettest day [mm day$^{-1}$]; (e) consecutive dry days [annual number of days]; and (f) effective radiative forcing (ERF) [W m$^{-2}$] for NTCF mitigation. The multi-model mean time series, and the corresponding trend estimated using a weighted least-squares regression, are included as thick black lines. The multi-model mean (MMM) trend, its significance and R$^2$ value, are also included, as is the 95% confidence interval (CI). Individual model mean trends are also included as defined by the legend.

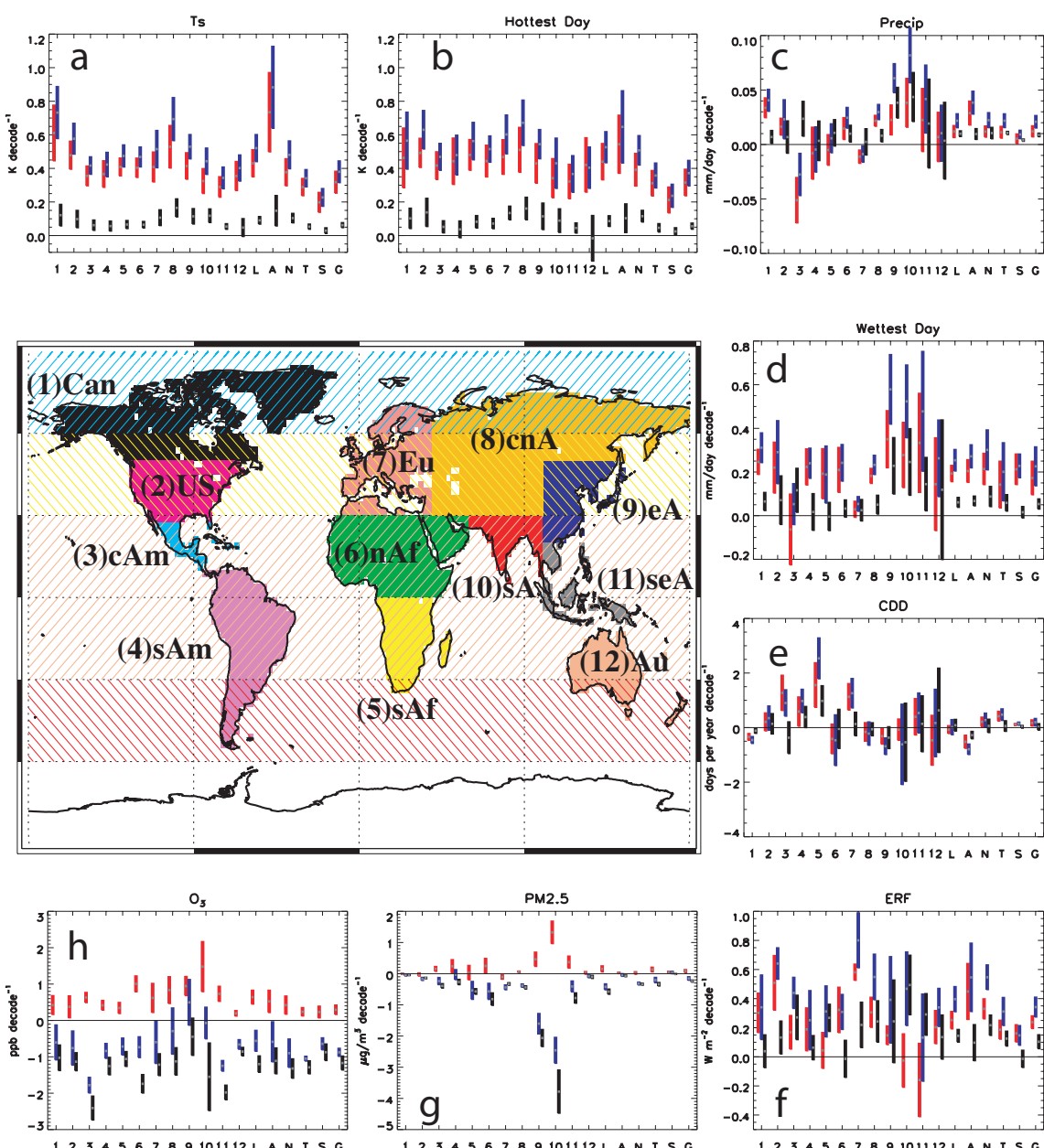

**Figure 5. Regional climate and air pollution responses to NTCF mitigation.** Bar plots show regional 2015-2055 trends in (a) surface temperature ($T_s$); (b) hottest day; (c) precipitation (Precip); (d) wettest day; (e) consecutive dry days (CDD); (f) effective radiative forcing (ERF); (g) surface particulate matter ($PM_{2.5}$) and (h) ozone ($O_3$) for weak (red) and strong (blue) air quality control, and NTCF mitigation (black). Bar center (gray horizontal line) shows the multimodel mean trend, estimated as the average of each model's mean trend. Bar length represents the 95% confidence interval, estimated as $2\sigma/\sqrt{n}$, where $\sigma$ is the standard deviation of the individual model mean trends and $n$ is the number of models. Center map shows the corresponding color coded world regions for each bar plot (as in Fig. 2). The average over these 12 land regions is abbreviated as "L". Also included is the Arctic ("A"; 60-90N; light blue hatched region); NH mid-latitudes ("N"; 30-60N; yellow hatched region); Tropics ("T"; 30S-30N; beige hatched region); SH mid-latitudes ("S"; 60-30S; red hatched region); and the global mean ("G"). Trend units are K decade$^{-1}$ for $T_s$ and hottest day; mm day$^{-1}$ decade$^{-1}$ for Precip and wettest day; $\mu$g m$^{-3}$ decade$^{-1}$ for $PM_{2.5}$; ppb decade$^{-1}$ for $O_3$; days per year decade$^{-1}$ for CDD; and W m$^{-2}$ decade$^{-1}$ for ERF.

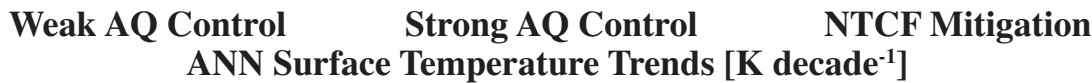

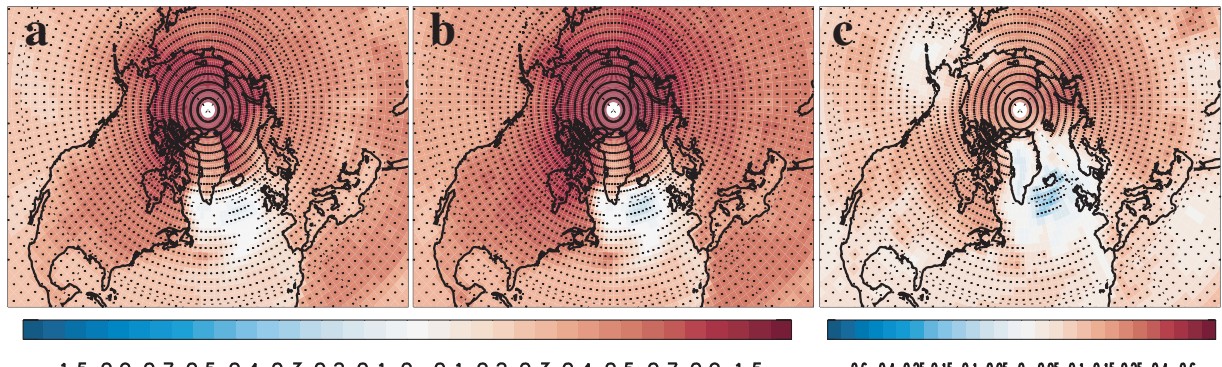

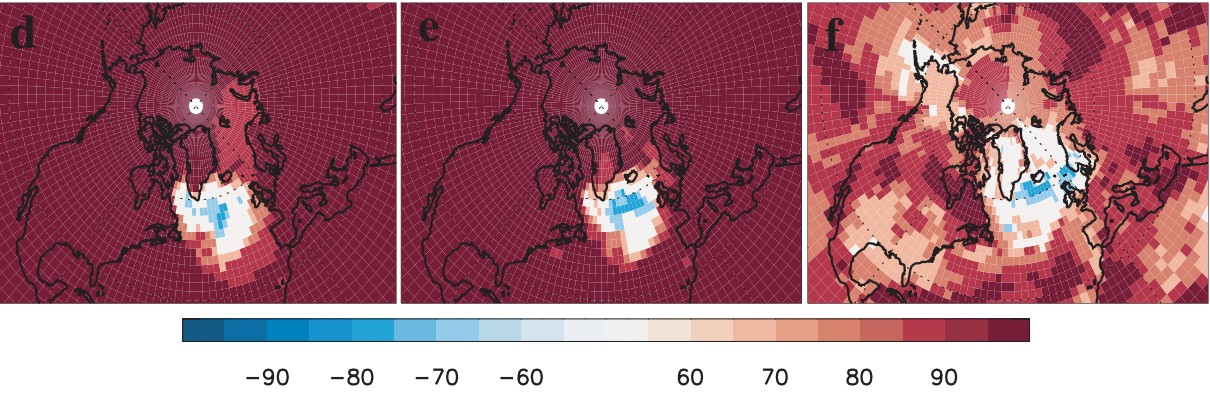

**Figure 6. 2015-2055 annual mean surface temperature trends and model trend realization agreement over the Arctic.** Surface temperature (a-c) trends [K decade$^{-1}$] and (d-f) model trend realization agreement [%] for (left panels) weak air quality control; (middle panels) strong air quality control and (right panels) NTCF mitigation. Stippling denotes trend significance at the 95% confidence level based on a standard $t$-test. Trend realization agreement represents the percentage of models that agree on the sign of the trend. Red colors indicate model agreement on a positive trend; blue colors indicate model agreement on a negative trend. White areas indicate lack of agreement on the sign of the trend.

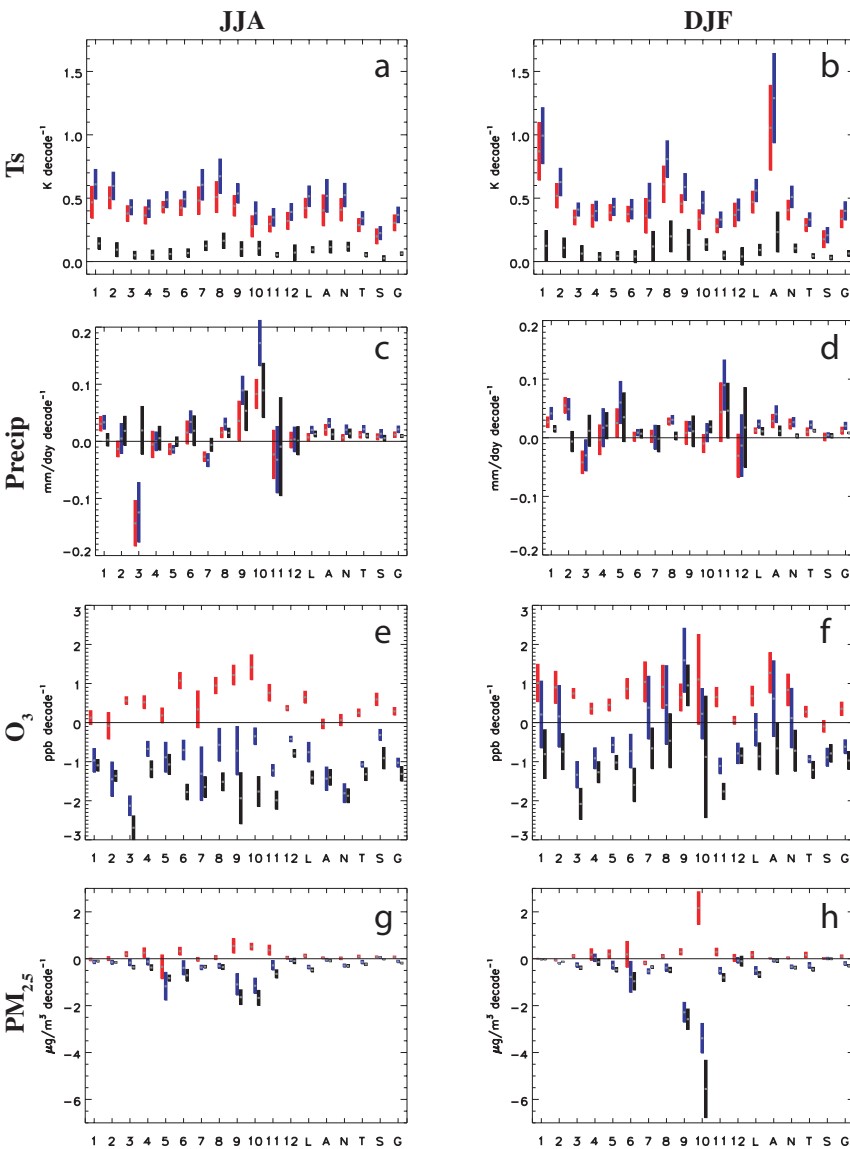

**Figure 7. Regional climate and air pollution seasonal responses to NTCF mitigation.** Bar plots show regional 2015-2055 June-July-August (JJA; left panels) and December-January-February (DJF; right panels) trends in (a-b) surface temperature ($T_s$); (c-d) precipitation (Precip); (e-f) surface ozone ($O_3$); and (g-h) surface particulate matter ($PM_{2.5}$) for weak (red) and strong (blue) air quality control, and NTCF mitigation (black). Bar center (gray horizontal line) shows the multimodel mean trend, estimated as the average of each model's mean trend. Bar length represents the 95% confidence interval, estimated as $2\sigma/\sqrt{n}$, where $\sigma$ is the standard deviation of the individual model mean trends and $n$ is the number of models. World regions are identical to those in Figure 5. Trend units are K decade$^{-1}$ for $T_s$; mm day$^{-1}$ decade$^{-1}$ for Precip; $\mu$g m$^{-3}$ decade$^{-1}$ for $PM_{2.5}$; and ppb decade$^{-1}$ for $O_3$.

**Table 1. Air pollution and climate responses to NTCF mitigation**. Annual mean 2015-2055 trends in surface particulate matter ($PM_{2.5}$), ozone ($O_3$), surface temperature ($T_s$), precipitation (Precip), hottest day, wettest day, consecutive dry days (CDD) and the effective radiative forcing (ERF) for NTCF mitigation. First set of numbers is the global mean trend; second set of numbers is the land-only trend. Trends significant at the 95% confidence level are denoted by bold font based on a $t$-test. Trend units are K decade$^{-1}$ for $T_s$ and hottest day; mm day$^{-1}$ decade$^{-1}$ for Precip and wettest day; $\mu$g m$^{-3}$ decade$^{-1}$ for $PM_{2.5}$; ppb decade$^{-1}$ for $O_3$; days per year decade$^{-1}$ for CDD; and W m$^{-2}$ decade$^{-1}$ for ERF. The first five models include both aerosol and ozone changes (Aer+O3 models); bottom four models include only aerosol changes (Aer models). MMM is the multi-model mean and the last row ("MMM Total") shows the total change over the entire 2015-2055 time period based on all models.

| | $PM_{2.5}$ | $O_3$ | $T_s$ | Precip | Hottest Day | Wettest day | CDD | ERF |
|---|---|---|---|---|---|---|---|---|
| **Aer+O3 Models** | | | | | | | | |
| UKESM1-0-LL | **−0.26/−0.67** | **−0.81/−0.81** | **0.07/0.09** | **0.011/0.017** | **0.05/0.05** | **0.055/0.100** | −0.17/−0.36 | **0.07**/0.02 |
| BCC-ESM1 | **−0.26/−0.51** | **−1.22/−1.13** | **0.09/0.14** | **0.009/0.010** | **0.09/0.14** | **0.111/0.095** | **0.25**/0.40 | −0.01/0.07 |
| GFDL-ESM4 | **−0.24/−0.57** | **−1.25/−1.26** | **0.07/0.08** | **0.011**/0.004 | n/a | n/a | n/a | 0.03/**0.13** |
| CESM2-WACCM | **−0.29/−0.78** | **−1.36/−1.39** | **0.08/0.10** | **0.010**/0.007 | n/a | 0.63/0.033 | −0.05/−0.06 | **0.16/0.23** |
| MRI-ESM2-0 | **−0.28/−0.58** | **−1.22/−1.42** | **0.04/0.07** | **0.010/0.007** | **0.06/0.09** | **0.058**/0.041 | 0.13/0.26 | **0.08/0.16** |
| MMM | **−0.26/−0.59** | **−1.19/−1.11** | **0.07/0.10** | **0.009**/0.012 | **0.07/0.11** | **0.064/0.067** | 0.12/0.07 | **0.07/0.12** |
| **Aer Models** | | | | | | | | |
| CNRM-ESM2-1 | **−0.15/−0.41** | n/a | **0.04/0.05** | **0.006/0.014** | **0.04/0.05** | **0.043/0.068** | 0.03/0.01 | **0.12/0.16** |
| MIROC6 | n/a | n/a | 0.02/0.04 | **0.004/0.010** | 0.03/0.06 | 0.027/0.024 | **0.13/0.41** | **0.11/0.21** |
| MPI-ESM1-2-HAM | **−0.23/−0.58** | n/a | **0.08/0.13** | **0.008**/0.010 | **0.08/0.12** | 0.016/0.041 | −0.14/−0.05 | **0.22/0.17** |
| NorESM2-LM | **−0.20/−0.48** | n/a | **0.08/0.11** | **0.010/0.012** | n/a | n/a | n/a | **0.16/0.16** |
| MMM | **−0.16/−0.44** | n/a | **0.06/0.09** | **0.005/0.009** | **0.04/0.10** | **0.039/0.061** | **0.07**/0.14 | **0.17/0.19** |
| **All Models** | | | | | | | | |
| MMM | **−0.23/−0.55** | **−1.19/−1.11** | **0.06/0.09** | **0.008/0.011** | **0.06/0.09** | **0.053/0.054** | 0.08/0.09 | **0.11/0.15** |
| MMM Total | −0.92/−2.20 | −4.76/−4.55 | 0.25/0.36 | 0.032/0.045 | 0.26/0.36 | 0.212/0.221 | 0.32/0.37 | 0.44/0.59 |