# Peer review of "Climate and air quality impacts due to mitigation of non-methane near-term climate forcers"

_Atmospheric Chemistry and Physics, 2019_

## Referee Comment (RC1) · Anonymous Referee #1 · 16 Feb 2020

General:

Allen et al. introduce results of the AerChemMIP project on the impact of air quality measures on climate. This is a large exercise and certainly worth publishing. However, I think there are major shortcomings. The most apparent is the style. The paper is written as a report, stating what has been done and what is the outcome. While this is, of course, an essential part of a paper, it should contain much more. It is less written as a scientific paper that should motivate chosen assumptions, extract main new massages from results, and discuss uncertainties e.g. wrt. to the chosen assumptions. This is largely missing. For example the main message "Our findings suggest that future policies that aggressively target non-methane NTCF reductions will improve air quality, but will lead to additional surface warming" is shown in the end

as being nothing new, but already covered by many other studies, as shown by the authors in line 345ff. So what is new? And this puts me actually in a difficult position, why should a paper be published which "just" confirms previous findings? I understand that IPCC deadlines have to be met, but more emphasis should be given to clearly describe what is new. More examples are given in the detailed comments below.

Major comments in addition to the writing style:

1) Structure: The method section is too short:

- More information on statistics should be given (see details below); Please explain why the multi-model trends are significant, although individual model trends are not. What trend model has been used? What exactly is tested?

- A motivation why exactly these climate/air quality/extreme indicators are chosen is missing;

- Part 3.1 is actually input to the study and should be moved from the results part to the method part.

2) Statistics: I have strong concerns how the statistics are interpreted. If a difference is not statistical significant, there is no basis in discussing them. Please remove all parts, which interpret statistically insignificant differences.

3) Discussion: How important are the choices made in the assumption section?

Detailed comments:

Abstract: "How future policies affect the abundance of NTCFs and their impact on climate and air quality remains uncertain." I am wondering whether this could be misunderstood in a way that for a given measure the impact remains uncertain. Most of the uncertainty comes from the uncertainty what measures will be taken, right?

l13 "similar increases" what means similar here? Can a extreme weather index be similar to a temperature increase of 0.24 K? or is even 0.34 K similar to 1.1%. Please

specify.

l16 "ozone reductions.": I think it would be helpful to include half a sentence explaining the relation between aerosols and ozone.

l20-21: I think the definition in Myhre et al 2013 is "We define 'near-term climate forcers' (NTCFs) as those compounds whose impact on climate occurs primarily within the first decade after their emission." It reads a little bit different from "that impact climate on relatively short time scales, typically within a few weeks to a decade after emission" Climate is defined on decadal timescales. To relate climate change to to weeks sounds weird. Concentration changes and RF can quickly react, but you started to discuss climatological changes in temperature and rain rates and those do not occur on weekly timescales. Please adapt the text.

l28 should it be "-2.0 to $-0.4$" ?

l34 shouldn't methane be mentioned here as well, since it is a precursor for ozone? I think you are referring to table 8.6 in Myhre et al. 2013. Their tropospheric ozone are a total ozone change and include effects from methane emissions.

l34-37: Here you change from a concentration perspective (ozone) to an emission perspective (methane). Please clarify this, otherwise it seems to be inconsistent and double counting methane ozone effects. Especially the wording "Similarly," should be revised, since the view is exactly not similar.

l42-44 please clarify the sentence. How can a change in radiation, i.e. in W/m2, be balanced by evaporation in units m/s.

l62 please clarify what you mean with "rapid". See also discussion above.

l91 You mean the scenarios you are employing. ...

Section 2.1: I think it would be nice to have a motivation included. Currently, it reads like a report or namelist setting. Why is the reference without climate policy? etc. this

should be motivated.

Please also include a table showing the changes in relevant emissions, such as aerosol compounds and ozone precursors for some well-chosen times, e.g. 2015, 2035, 2055; or decadal? I think it is important to see the changes.

l120 I find the abbreviation misleading. "lowAER" and "lowAERO3" are model group names. "low", however, is not referring to the models, but to the scenario, right? and at some point I though "AERO3" is the "AEROsol Group 3" and not aerosol-ozone. What about "Only-Aer" and "Aer-O3"; or "Aer+O3" ?

l135 Please include what kind of linear model you are using $y(t)=a+bt+err$ or $y(t)=b(t-2035)+err$ ? Are you fitting one or two parameters? Often as a measure for the fitting quality the $R^2$ value or adjusted $R^2$ value is used. Why not here? I do not understand how the trend is tested. Are the individual model results fitted and then tested whether the mean trend is representing the range of models correctly? (At least the caption of Figure 7 might indicate something like this). How the statistics are treated is very important for the interpretation. Please include a thorough discussion here.

l159 Also here a motivation is missing. I understand that extreme values are important. But why is the max temperature chosen? Isn't that a statistically very difficult quantity, even among extreme value statistics? Why not using number of hot days, i.e. over 30°C? This also concentrates on extremes, but includes a whole tail of a pdf (or estimated pdf).

l162 please also add the respective time frame. Are you averaging over 10 or 30 years?

l165ff: I would have expected this part in the scenario section. Please consider to move it there, since this is not a result from your paper, right? And then ignore my comment on the table (see above) ...

l167: Why is there a $CO_2$ emission change at all, if you are considering NTCF changes only? Please explain. I don't think that this is a problem, but currently and certainly it

confuses me.

l192: Why are you discussing methane emission changes, if those are not relevant?

Figure 2: Trends are calculated as (2055-2015)/4 or with the regression method discussed in Section 2?

l 205: Please comment if the trends of the individual models are statistical significant. I miss a mathematical/statistical explanation in combination with a motivation why to test the multi-model mean and not, whether mean trend is significant wth respect to the variation in trends of the individual models.

l 206: For the regional trend an uncertainty range is given. Why not here?

l213-l214: If a result is not statistical significant, there is no point in interpreting the result. Please delete the sentence.

l223: Keep in mind that the change was not statistical significant; so the results may not be inconsistent, but only noise. Please revise the discussion, based on what is inconsistent on a statistical significant basis.

l 246: "Slightly larger (but not statistically significant) "; if not statistically significant, then they are not slightly larger! Please respect the statistics.

l263-264: Please rephrase the sentence. I agree with the content, however, the formulation, starting with "however" suggests that there is either a shortcoming or something unexpected, etc. As the authors state this is by no means a surprise nor limitation of the aforementioned.

l262 I think somewhere it should made clear that a part of the warming is a reduced cooling from SO2 reduction and O3 reductions, right?

l284 Is there some relation to the monsoon tipping points?

Section 3.4: What about MAM/SON? Discussed are winter/summer differences. How-

ever, "Seasons" would imply more than that. I suggest to, at least, mention a general trend for MAM/SON and add the same figures in a supplement.

l339-342: This is important: see also above. If the difference is not statistical significant, there is no point in discussing or even highlighting it in the summary. Please remove this part!

l359: It might be worth mentioning reduced warming, i.e. a net cooling. To avoid confusion about weather CH4 itself has a cooling contribution.

---

## Referee Comment (RC2) · Anonymous Referee #2 · 4 Mar 2020

Allen et al. use model output from the AerChemMIP intercomparison project to evaluate 2015-2055 changes in climate variables associated with two future air quality control scenarios. By comparing a "weak" policy scenario to a "strong" policy scenario, they show increasing trends in temperature and precipitation over the period that are driven by near-term climate forcers (ozone and aerosols), suggesting a climate penalty associated with air quality improvements.

The manuscript is generally well written and well structured and makes good use of the AerChemMIP simulations. It addresses an important question that is well suited to the scope of ACP. I do have a few concerns about the statistics and a few more minor comments and suggestions, discussed below.

GENERAL COMMENTS

[Figure]

1. The trends have been calculated using least squares regression. There is very little information on exactly how that was implemented, so from my reading it does not appear that this is a weighted least squares regression, or that the uncertainties have been accounted for in any other way. This is concerning because, looking at Fig. 3 for example, there is a large amount of variability in the individual models that are used to construct the multi-model means. I am not convinced by the robustness of some of the reported trends in the multi-model mean, or that they are truly statistically significant as stated. The multi-model mean trend calculations should be performed using a method that accounts for variability/uncertainty in the mean (e.g., weighted least squares, but there are other options) before the paper is publishable in ACP. In addition, some discussion of the method used and the influence of the variability/uncertainty is warranted.

2. For the global trends in climate variables, it would help to contextualise the values associated with NTCFs by also providing the trends from the two individual scenarios (or at least from the one with weak NTCF control, as the other can be determined from the difference trends provided). Without this, it's hard to tell how important the NTCF climate penalty is. I note that this is done in the figures for the regional trends, but not for the global trends. I would strongly encourage the authors to add these in some form (for example, a figure in the SI equivalent to Fig. 3).

3. The manuscript is very well structured and quite well written, but the heavy use of acronyms and technical identifiers (e.g., SSP3-7.0-lowNTCF, lowAERO3, etc.) makes it harder to read & follow than it needs to be. I would encourage the authors to simplify this wherever possible and then use a consistent, easy to interpret nomenclature throughout. For example, frequently the two scenarios are referred to as strong and weak air quality control, and these are much easier to interpret than SSP3-7.0 and SSP3-7.0-lowNTCF. I would suggest strong and weak air quality control could replace SSP3-7.0 and SSP3-7.0-lowNTCF everywhere, in particular in figure legends and captions where the reader may not be referring back to the text. Similarly, NTCF mitigation is easier to interpret than SSP3-7.0-lowNTCFSSP3-7.0.

4. The manuscript cites a lot of "in prep" and "submitted" papers. In most cases, these are cited as part of long lists of other references, so they aren't really needed to make the points. If these are not at least in ACPD by the time of publication, they should not be included in the citation lists (except in cases where they are the only publications available to back-up the point).

SPECIFIC COMMENTS

L30: Does the net radiative effect here refer just to OC or to BC+OC? Please rephrase to clarify.

L59: Is this newer estimate of mortality for all air pollution or outdoor ambient only? Please rephrase to clarify.

L90-108: This information would benefit from being summarised in a table listing the scenarios and some of the relevant information (e.g. air quality controls weak/strong, ozone and aerosols high/low, CH4 high/high, etc.) to make it easier for the reader to synthesise.

L120-122: I find this a bit confusing. What is the difference between CESM2 and CESM2-WACCM in this case? Is it the aerosol treatment? And if they are basically the same model, is it fair to include them as two separate data points in the multi-model means?

L141-144: So nitrate aerosol was not included in PM2.5 at all, even for the models that do include it? It would be nice to see how much uncertainty this adds, given nitrate can be an important component of aerosol loading in some regions. I'd suggest adding a version of the PM2.5 figures including nitrate to the supplement, and a brief discussion of the impacts of excluding nitrate either in the main text or in the supplement.

L156: Is there a reference for these ground-based observations? Or is this the same GASSP observations mentioned above? If the latter, please state explicitly in the text.

L172-L180: This is confusing when paired with the figure. It is completely legitimate to

not include the differences in CH4 pathways for this work, but anyone skimming quickly and focusing on the figures will miss that point. In my opinion, Figure 1 should only show what was used in this work, not scenarios that are not used here. I strongly encourage the authors to remove the SSP3-7.0-lowNTCF (right?) and difference lines from Figure 1. The comparison between the scenarios can be moved to the supplement if the authors feel it is important to include.

L181-187: Similarly, I don't think this discussion belongs here. It is the first section of the results, yet it is mostly discussing what is not done in this work. I would suggest this could be removed entirely, or moved to the supplement or to the conclusions as part of a discussion of what future work should be done to build on what the authors have done here. Sect. 3.2 and Figs 3-4: Generally speaking, is the changes in atmospheric composition (aerosols and ozone) that are driving the changes in climate. Thus it seems a bit odd to show and discuss the changes in climate variables BEFORE the changes in composition (ozone, PM2.5). I would suggest restructuring such that Fig. 3 comes before Fig. 4, with the text order changed to match. (I note this is already the order used in the abstract and conclusions).

L204-205: Unless you rename & define the scenarios in the methods as discussed above, please clarify how "under NTCF mitigation" is defined here (I understand that it is the difference between the two scenarios, but that wasn't clear to me on first read).

L211-218, L223-228 (and elsewhere): Much is made of the difference between the lowAER and lowAERO3 outcomes. Given that one of these only includes 3 models and the manuscript states explicitly that the difference is not significant, it is not justifiable to be interpreting this as a result. This appears to me to be over-interpretation of noise, and I would suggest this discussion be removed before publication in ACP.

L233: This land-only result appears to be insignificant for 75% of the models (including those that show increases) and so this statement should be removed or qualified.

L238: CDD does not show a statistically significant increase in the overall MMM (or

in the subset MMM or in any of the individual models bar one) – therefore should be removed from this sentence.

L255-256: Is the land-only warming pattern shown anywhere? Is the land-only warming weaker or stronger than the overall warming? If there is a difference, it would be useful to see an equivalent figure in the supplement. (And if there is not a difference, it's not clear why this is discussed separately.)

L264: "...forcing and response do not need to occur in the same regions." Can this be explained a bit more?

L269-271: Do I understand Fig 6 bottom panel correctly that models don't agree about this feature? If so that would be worth stating in this discussion

L307: For a discussion of seasonal patterns to make sense, consideration should be given to the different seasonalities of the two hemispheres. Figure 7 should either be separated or at least ordered/demarcated by hemisphere – I'd suggest NH extratropics, tropics, and SH tropics.

L370: Why is one model listed explicitly when all (including that model) are available from the same location? Also please spell out ESGF here and provide a link or doi.

Figs 2, 5,7: regional legend labels on x-axis are impossible to read because they are so small. Perhaps give each region a number instead? Or include some other sort of key to make this clearer?

Fig 3 caption (and elsewhere): Does "hottest day" refer to "surface temperature on hottest day"? Similarly for wettest day? Please clarify somewhere.

Fig 3 caption: It seems the thin coloured lines show the trends for the individual models, but this has not been explicitly stated in the caption. Please update caption to clarify.

Figs 3, 4, 5: why are different units used for the trends in the precip variables (mm/day vs. %) in the global and regional trend figures? Same question for PM2.5 and O3. Can

these be standardised to more easily compare?

Fig 6d-f: these plots are not currently discussed in the text and therefore should perhaps move to the supplement (or be mentioned in the text)

Table 1: I found this table hard to understand while trying to refer to it while reading the text. A few suggestions to improve the clarity. (1) Add lowAER and lowAERO3 identifiers above the list of relevant models in each sub-section so it's easy to see which group is which. (2) If text and figures are re-ordered as suggested above, move PM2.5 and O3 columns to be left-most, followed by the climate variables. (3) Move the three "MMM total" lines either to a separate part of the table or (preferably) to a new table altogether as the numbers aren't comparable to the lines above/below which makes it difficult to interpret (and already a lot to interpret in the table!). (4) For the lowAER models' O3 response, replace 0.0 with n/a since these values are not included in the Overall MMM calculation (as is, looks like the overall will be an average of the lowAERO3 values and three zeros).

TECHNICAL COMMENTS

L139: "were are" –> "we are"

L357: "complex" –> "complexity"

Fig 2 caption: "astriks" –> "asterisks"

---

## Author Comment (AC1) · 13 May 2020

**Response to Reviewer #1**

We thank Reviewer #1 for evaluating our manuscript. Below, we list our responses to each comment (in blue). We first note that all analyses have been updated based on current model availability. This includes the addition of more realizations and/or climate variables to the models previously used (e.g., we now have MIROC daily data and 2 additional CESM2-WACCM simulations, etc.). We have also added two additional models to the analysis: NorESM2-LM and UKESM1-0-LL. Our overall results and conclusions remain unchanged.

**Reviewer #1**

General:

Allen et al. introduce results of the AerChemMIP project on the impact of air quality measures on climate. This is a large exercise and certainly worth publishing. However, I think there are major shortcomings. The most apparent is the style. The paper is written as a report, stating what has been done and what is the outcome. While this is, of course, an essential part of a paper, it should contain much more. It is less written as a scientific paper that should motivate chosen assumptions, extract main new massages from results, and discuss uncertainties e.g. wrt. to the chosen assumptions. This is largely missing. For example the main message "Our findings suggest that future policies that aggressively target non-methane NTCF reductions will improve air quality, but will lead to additional surface warming" is shown in the end as being nothing new, but already covered by many other studies, as shown by the authors in line 345ff. So what is new? And this puts me actually in a difficult position, why should a paper be published which "just" confirms previous findings? I understand that IPCC deadlines have to be met, but more emphasis should be given to clearly describe what is new. More examples are given in the detailed comments below.

We have attempted to improve the writing, by placing more emphasis on motivation and assumptions. Although our main results support prior studies, given the sophistication of the models used here, as well as the relatively large number of models, we suggest that this work is the most comprehensive analysis on this topic to date. Overall, the structure of the manuscript is similar to the original submission. Reviewer #2 states: "…the manuscript is generally well written and well-structured and makes good use of the AerChemMIP simulations".

Major comments in addition to the writing style:
1) Structure: The method section is too short:

- More information on statistics should be given (see details below); Please explain why the multi-model trends are significant, although individual model trends are not. What trend model has been used? What exactly is tested?

More information on the statistics are included. Models with multiple realizations are first averaged to form the model mean. Individual model mean trends are calculated using least squares regression, and the corresponding trend significance is based on a two-tailed Student's t-test, where the null hypothesis of a zero regression slope is evaluated. Autocorrelation of the

time series is also accounted for by using the effective sample size: $n \times (1 - r_1)/(1 + r_1)$, where $n$ is the number of years and $r_1$ is the lag-1 autocorrelation coefficient.

MMM trends and their significance are estimated in two ways, and both methods yield similar results. In the first approach, the overall multi-model mean time series is calculated as the mean over each model mean (i.e., each model has the same weight), and a similar procedure as above is used to determine the significance of the multi-model mean trend. However, we now use a weighted least squares regression (as suggested by Reviewer #2). Each value in the multi-model mean time series is weighted by $\frac{1}{\sigma_m^2}$, where $\sigma_m$ is the standard deviation across models.

We note that this does not change any of our results. For example, the global annual mean multi-model surface temperature trend changes from 0.062 without weighting to 0.060 K/decade with weighting, and both are significant at the 99% confidence level. Weighting the regression introduces negligible changes in our other climate and air quality trends.

In Figure 3, there was actually quite a bit of similarity in individual global mean model trends. All but one individual model surface temperature trend in Figure 3 was significant (MIROC6 the lone exception). Furthermore, all individual model precipitation, Hottest Day, PM2.5 an Ozone global mean trends were significant. Weaker results existed for the Wettest Day, and in particular, Consecutive Dry Days. It is therefore not surprising the multi-model mean trends were also significant. Using the weighted regression approach, we get similar conclusions.

To summarize, we now use a weighted least squares regression for the multi-model mean trends, and we get similar results as before.

In addition to estimating the magnitude and significance of the multi-model mean trend as just described, we also evaluate the multi-model mean trend and its significance relative to the individual model mean trends (e.g., Figure 5). Here, the MMM trend is estimated as the average of each model mean trend, and its uncertainty is estimated as plus/minus twice the standard error (i.e., the 95% confidence interval). This is calculated as: $2 \times \sigma / \sqrt{nm}$, where $\sigma$ is the standard deviation of model mean trends and $nm$ is the number of models. If this confidence interval does not include zero, then the multi-model mean trend is significant at the 95% confidence level.

The corresponding 95% confidence intervals are now included in each of the global time series plots. As is the $R^2$ value of the multi-model mean trend.

We have also added the multi-model global mean trend (and others, including NH mid-latitudes, Tropics, SH mid-latitudes) and its uncertainty to the bar graphs in Figure 5.

- A motivation why exactly these climate/air quality/extreme indicators are chosen is missing;

As discussed in the Introduction, both PM2.5 and ozone are commonly used indicators of air quality. Both have been associated with adverse human health impacts. Surface temperature and precipitation are analyzed as these are arguably two of the most important climate variables. Changes in surface temperature are particularly relevant in the context of climate mitigation, as

the goal of the Paris Agreement is to keep the increase in global mean surface temperature to well below 2°C above preindustrial values. Precipitation, and fresh water resources in general, are important to both human society and ecosystems. Perhaps more important than changes in the mean of a climate variable are changes in its extremes. Heat waves, for example, are a major cause of weather related fatalities. We focus here on the hottest and wettest day, as well as consecutive dry days, as these are frequently used extreme temperature and precipitation indices (e.g., Donat et al., 2013a,b). Furthermore, prior observational analyses have shown significant changes in all three quantities over the latter half of the 20[th] century. This information has been added.

- Part 3.1 is actually input to the study and should be moved from the results part to the method part.

Moved to the Methods section.

2) Statistics: I have strong concerns how the statistics are interpreted. If a difference is not statistical significant, there is no basis in discussing them. Please remove all parts, which interpret statistically insignificant differences.

These have been removed.

3) Discussion: How important are the choices made in the assumption section?

We assume the "assumption section" pertains to the future emission pathways used here. We have added more information on the assumptions pertaining to the two future emissions pathways analyzed here. Basically, our analysis assumes that NTCF policies can be enacted in the absence of GHG related climate policies (e.g., SSP1's air pollutant legislation and technological progress can be achieved in the SSP3 world). Furthermore, our results likely represent an upper bound, since our baseline/reference scenario lacks climate policy and has the highest levels of NTCFs. This has been clarified in the Future Scenarios Section.

In the Conclusion Section, we discuss the implications of the assumptions made in the weak and strong air quality control pathways used in this analysis. It is not possible to formally quantify these assumptions, as different NTCF mitigation simulations were not performed by AerChemMIP. We have added clarification in the conclusions:

Our simulations, however, do not account for CO2 reductions, implying the importance of simultaneous reductions in both CO2 and NTCFs. We note that it is difficult to reduce only the NTCF emissions while keeping CO2 emissions fixed (since there are co-emitted species, including SO2). If CO2 emissions are simultaneously reduced along with NTCFs, then the increase in global surface temperature and precipitation found here will be muted.

Detailed comments:

Abstract: "How future policies affect the abundance of NTCFs and their impact on climate and air quality remains uncertain." I am wondering whether this could be misunderstood in a way that for a given measure the impact remains uncertain. Most of the uncertainty comes from the uncertainty what measures will be taken, right?

Future climate and air quality are uncertain for two reasons. There is uncertainty due to the emissions pathway, and there is uncertainty in the corresponding climate response. Past IPCC reports have shown that both uncertainties are approximately of the same magnitude in the context of climate change. The latter uncertainty is due to uncertainty in climate sensitivity (e.g., 1.5-4.5 K per 2xCO2). As an aside, CMIP6 models tend to have a higher climate sensitivity than CMIP5 models, which has been related to clouds (e.g., Zelinka et al., 2020). Nonetheless, we have attempted to clarify this sentence, since the larger uncertainty for a given pathway is the climate response.

l13 "similar increases" what means similar here? Can a extreme weather index be similar to a temperature increase of 0.24 K? or is even 0.34 K similar to 1.1%. Please specify.

Re-worded.

l16 "ozone reductions.": I think it would be helpful to include half a sentence explaining the relation between aerosols and ozone.

We have added information in the Introduction. "…reductions in some precursor gases such as NOx and VOCs impact both ozone and aerosols (and perhaps CH4). Reductions in NOx, for example, will promote cooling due to reduced tropospheric ozone, but the impact on CH4 lifetime and aerosol formation will likely promote overall warming"

l20-21: I think the definition in Myhre et al 2013 is "We define 'near-term climate forcers'(NTCFs) as those compounds whose impact on climate occurs primarily within the first decade after their emission." It reads a little bit different from "that impact climate on relatively short time scales, typically within a few weeks to a decade after emission". Climate is defined on decadal timescales. To relate climate change to to weeks sounds weird. Concentration changes and RF can quickly react, but you started to discuss climatological changes in temperature and rain rates and those do not occur on weekly timescales. Please adapt the text.

We have adopted the reviewer's verbiage.

l28 should it be "-2.0 to−0.4" ?

Re-ordered.

l34 shouldn't methane be mentioned here as well, since it is a precursor for ozone? I think you are referring to table 8.6 in Myhre et al. 2013. Their tropospheric ozone area total ozone change and include effects from methane emissions.

Methane added here.

l34-37: Here you change from a concentration perspective (ozone) to an emission perspective (methane). Please clarify this, otherwise it seems to be inconsistent and double counting methane ozone effects. Especially the wording "Similarly," should be revised, since the view is exactly not similar.

Modified to concentration perspective.

l42-44 please clarify the sentence. How can a change in radiation, i.e. in W/m2, be balanced by evaporation in units m/s.

Changed evaporation to latent cooling.

l62 please clarify what you mean with "rapid". See also discussion above.

Clarified.  Added "decadal".

l91 You mean the scenarios you are employing. ...

Added "Used here".

Section 2.1: I think it would be nice to have a motivation included. Currently, it reads like a report or namelist setting. Why is the reference without climate policy? etc. this should be motivated.

We have added motivation, assumptions, and clarity. Our analysis assumes that NTCF policies can be enacted in the absence of GHG related climate policies (e.g., SSP1's air pollutant legislation and technological progress can be achieved in the SSP3 world).  Furthermore, our results likely represent an upper bound, since our baseline/reference scenario lacks climate policy and has the highest levels of NTCFs (i.e., to detect the largest signal, the reference is without climate policy).

Please also include a table showing the changes in relevant emissions, such as aerosol compounds and ozone precursors for some well-chosen times, e.g. 2015, 2035, 2055;or decadal? I think it is important to see the changes.

Figure 2 already shows changes in emissions by region, as well as over land.

l120 I find the abbreviation misleading. "lowAER" and "lowAERO3" are model group names. "low", however, is not referring to the models, but to the scenario, right? and at some point I though "AERO3" is the "AEROsol Group 3" and not aerosol-ozone. What about "Only-Aer" and "Aer-O3"; or "Aer+O3" ?

Changed abbreviations to "Aer" and "Aer+O3"

l135 Please include what kind of linear model you are using y(t)=a+bt+err or y(t)=b(t-2035)+err ? Are you fitting one or two parameters? Often as a measure for the fitting quality the R^2 value or adjusted R^2 value is used. Why not here? I do not understand how the trend is tested. Are the individual model results fitted and then tested whether the mean trend is representing the range of models correctly? (At least the caption of Figure 7 might indicate something like this). How the statistics are treated is very important for the interpretation. Please include a thorough discussion here.

Our response to the concern over statistics is located above.

l159 Also here a motivation is missing. I understand that extreme values are important. But why is the max temperature chosen? Isn't that a statistically very difficult quantity, even among extreme value statistics? Why not using number of hot days, i.e.over 30◦C? This also concentrates on extremes, but includes a whole tail of a pdf (or estimated pdf).

The Hottest Day ("TXx") is chosen as it is a commonly used extreme temperature measure.   See for example, Donat et al. (2013 a,b).  As with other extreme temperature indices, significant increases in TXx were found (1951-2011) in two different data sets, GHCNDEX and HadGHCND.  We also find significant TXx trends in the simulations analyzed here.

l162 please also add the respective time frame. Are you averaging over 10 or 30 years?

We are not sure what time frame the reviewer refers to.  L162 states "Climate extremes are calculated at each grid box and then spatially averaged."   There is no decadal averaging.

The hottest day (monthly maximum value of daily maximum temperature) and the wettest day (monthly maximum 1-day precipitation) are estimated for each month, and then averaged to obtain annual means.   Consecutive dry days (CDD), defined as the maximum annual number of consecutive days with precipitation <1 mm/day, are estimated for each year.

l165ff: I would have expected this part in the scenario section. Please consider to move it there, since this is not a result from your paper, right? And then ignore my comment on the table (see above) ...

Moved to scenario section.

l167: Why is there a CO2 emission change at all, if you are considering NTCF changes only? Please explain. I don't think that this is a problem, but currently and certainly it confuses me.

Yes, there are small differences in CO2 emissions between the two scenarios. Methane reductions generate emissions abatement costs, which changes industrial outputs in all productin sectors and household consumption (Gidden et al., 2019).  Energy consumption and CO2

emissions in all sectors are thus affected, which causes small differences between SSP3-7.0 and SSP3-LowNTCF.

However, AerChemMIP simulations use the same CO2 emissions, based on SSP3-7.0 (as with methane).  This has been clarified in the revision.   We have also removed the SSP3-7.0-lowNTCF CO2 (and CH4) emissions from the plots, to avoid unnecessary confusion.

l192: Why are you discussing methane emission changes, if those are not relevant?

Deleted discussion on methane emission changes.

Figure 2: Trends are calculated as (2055-2015)/4 or with the regression method discussed in Section 2?

Emission trends are calculated using the same method as above.  Trends are based on a least squares regression, with significance based on a two-tailed Student's t-test.  We note that the emissions data is decadal after 2015, with monthly values for the year 2015, 2020, 2030, 2040, 2050, 2060, etc.  We estimate the emissions in 2055 as the mean of the emissions in 2050 and 2060 at each grid box.  We have added this information.

l 205: Please comment if the trends of the individual models are statistical significant. I miss a mathematical/statistical explanation in combination with a motivation why to test the multi-model mean and not, whether mean trend is significant wth respect to the variation in trends of the individual models.

Table 1 lists the trend (and whether it is significant at the 95% confidence level) for each model.  All but one model yields a significant global mean increase in surface temperature.  This has been clarified.

We initially did not evaluate the significance of the multi-model mean trend, relative to the individual model trends, for the global mean quantities.  We did this for the regional trends (e.g., Figure 5).  We do note that Figure 5 included land only averages.  Nonetheless, we have now performed this analysis for the global mean quantities.  The 95% confidence interval is now included in the global time series plots.  We have also added the multi-model global mean trend and its uncertainty to the bar graphs in Figure 5 (and additional latitudinal bands).

l 206: For the regional trend an uncertainty range is given. Why not here?

Yes, we have now added this analysis for the global trends.  The corresponding 95% confidence intervals are now included in each of the global time series plots.  We have also added the multi-model global mean trend and its uncertainty to the bar graphs in Figure 5.

l213-l214: If a result is not statistical significant, there is no point in interpreting the result. Please delete the sentence.

We agree, and this statement has been modified.  It is still interesting that similar (i.e., non-significant differences) global mean surface air temperature trends occur in Aer and Aer+O3 models.  We acknowledge that this could be due to several factors, but one interpretation is weak surface cooling due to reductions in ozone.

l223: Keep in mind that the change was not statistical significant; so the results may not be inconsistent, but only noise. Please revise the discussion, based on what is inconsistent on a statistical significant basis.

Discussion edited and revised.  As with surface temperature, non-significant differences between Aer and Aer+O3 ERF trends exist.

l 246: "Slightly larger (but not statistically significant) "; if not statistically significant, then they are not slightly larger! Please respect the statistics.

Edited.  Similar increases occur in both Aer and Aer+O3 models.

l263-264: Please rephrase the sentence. I agree with the content, however, the formulation, starting with "however" suggests that there is either a shortcoming or something unexpected, etc. As the authors state this is by no means a surprise nor limitation of the aforementioned.

Deleted "However"

l262 I think somewhere it should made clear that a part of the warming is a reduced cooling from SO2 reduction and O3 reductions, right?

Yes, based on the information presented in the Introduction (e.g., radiative forcing), SO2 reductions will warm.  But O3 reductions will cool.  This has been clarified.

From the Introduction: Thus, reductions in some NTCFs, including non-absorbing aerosols, will warm the climate system, whereas reductions in other NTCFs, including absorbing aerosols, tropospheric ozone, and methane will cool the climate system.

l284 Is there some relation to the monsoon tipping points?

L284 states: "Furthermore, in agreement with prior studies, precipitation increases in several monsoon regions, including east Africa, south Asia, and east Asia."  Thus, unlike the buildup of aerosols over the 20[th] century, future NTCF mitigation and continued increases in GHGs will likely accelerate the monsoons.  Not exactly sure what the Reviewer wants us to change here.

Section 3.4: What about MAM/SON? Discussed are winter/summer differences. However, "Seasons" would imply more than that. I suggest to, at least, mention a general trend for MAM/SON and add the same figures in a supplement.

Added general trends in MAM/SON seasons.  Added MAM/SON plots to supplement.

l339-342: This is important: see also above. If the difference is not statistical significant, there is no point in discussing or even highlighting it in the summary. Please remove this part!

We have rephrased. The lack of significant trend differences in Aer and Aer+O3 models is interesting. We acknowledge that this could be related to several factors. But one possible interpretation is weak surface cooling due to reductions in ozone. We feel as if our ability to compare Aer and Aer+O3 models is one of the novelties of this study. But again, we acknowledge this comes with caveats.

l359: It might be worth mentioning reduced warming, i.e. a net cooling. To avoid confusion about weather CH4 itself has a cooling contribution.

Added "net cooling" here.

---

## Author Comment (AC2) · 13 May 2020

**Response to Reviewer #2**

We thank Reviewer #2 for evaluating our manuscript. Below, we list our responses to each comment (in blue). We first note that all analyses have been updated based on current model availability. This includes the addition of more realizations and/or climate variables to the models previously used (e.g., we now have MIROC daily data and 2 additional CESM2-WACCM simulations, etc.). We have also added two additional models to the analysis: NorESM2-LM and UKESM1-0-LL. Our overall results and conclusions remain unchanged.

**Reviewer #2**

Allen et al. use model output from the AerChemMIP intercomparison project to evaluate 2015-2055 changes in climate variables associated with two future air quality control scenarios. By comparing a "weak" policy scenario to a "strong" policy scenario, they show increasing trends in temperature and precipitation over the period that are driven by near-term climate forcers (ozone and aerosols), suggesting a climate penalty associated with air quality improvements. The manuscript is generally well written and well structured and makes good use of the AerChemMIP simulations. It addresses an important question that is well suited to the scope of ACP. I do have a few concerns about the statistics and a few more minor comments and suggestions, discussed below.

GENERAL COMMENTS

1. The trends have been calculated using least squares regression. There is very little information on exactly how that was implemented, so from my reading it does not appear that this is a weighted least squares regression, or that the uncertainties have been accounted for in any other way. This is concerning because, looking at Fig. 3 for example, there is a large amount of variability in the individual models that are used to construct the multi-model means. I am not convinced by the robustness of some of the reported trends in the multi-model mean, or that they are truly statistically significant as stated. The multi-model mean trend calculations should be performed using a method that accounts for variability/uncertainty in the mean (e.g., weighted least squares, but there are other options) before the paper is publishable in ACP. In addition, some discussion of the method used and the influence of the variability/uncertainty is warranted.

More information on the statistics are included. Models with multiple realizations are first averaged to form the model mean. Individual model mean trends are calculated using least squares regression, and the corresponding trend significance is based on a two-tailed Student's t-test, where the null hypothesis of a zero regression slope is evaluated. Autocorrelation of the time series is also accounted for by using the effective sample size: $n \times (1 - r_1)/(1 + r_1)$, where $n$ is the number of years and $r_1$ is the lag-1 autocorrelation coefficient.

MMM trends and their significance are estimated in two ways, and both methods yield similar results. In the first approach, the overall multi-model mean time series is calculated as the mean over each model mean (i.e., each model has the same weight), and a similar procedure as above is used to determine the significance of the multi-model mean trend. However, we now use a

weighted least squares regression (as suggested by the Reviewer).  Each value in the multi-model mean time series is weighted by $\frac{1}{\sigma_m^2}$, where $\sigma_m$ is the standard deviation across models.

We note that this does not change any of our results.  For example, the global annual mean multi-model surface temperature trend changes from 0.062 without weighting to 0.060 K/decade with weighting, and both are significant at the 99% confidence level.  Weighting the regression introduces negligible changes in our other climate and air quality trends.

In Figure 3, there was actually quite a bit of similarity in individual global mean model trends.  All but one individual model surface temperature trend in Figure 3 was significant (MIROC6 the lone exception).  Furthermore, all individual model precipitation, Hottest Day, PM2.5 an Ozone global mean trends were significant.  Weaker results existed for the Wettest Day, and in particular, Consecutive Dry Days.  It is therefore not surprising the multi-model mean trends were also significant. Using the weighted regression approach, we get similar conclusions.

To summarize, we now use a weighted least squares regression for the multi-model mean trends, and we get similar results as before.

In addition to estimating the magnitude and significance of the multi-model mean trend as just described, we also evaluate the multi-model mean trend and its significance relative to the individual model mean trends (e.g., Figure 5). Here, the MMM trend is estimated as the average of each model mean trend, and its uncertainty is estimated as plus/minus twice the standard error (i.e., the 95% confidence interval).  This is calculated as: $2 \times \sigma / \sqrt{nm}$, where $\sigma$ is the standard deviation of model mean trends and $nm$ is the number of models.  If this confidence interval does not include zero, then the multi-model mean trend is significant at the 95% confidence level.

The corresponding 95% confidence intervals are now included in each of the global time series plots.  As is the R^2 value of the multi-model mean trend.

We have also added the multi-model global mean trend (and others, including NH mid-latitudes, Tropics, SH mid-latitudes) and its uncertainty to the bar graphs in Figure 5.

2. For the global trends in climate variables, it would help to contextualise the values associated with NTCFs by also providing the trends from the two individual scenarios(or at least from the one with weak NTCF control, as the other can be determined from the difference trends provided). Without this, it's hard to tell how important the NTCF climate penalty is. I note that this is done in the figures for the regional trends, but not for the global trends. I would strongly encourage the authors to add these in some form (for example, a figure in the SI equivalent to Fig. 3).

These have been added to Figure 5.  The last set of bars (labeled "GL") now show the global mean trends for SSP3-7.0, SSP3-7.0-lowNTCF and their difference.   Also included is the corresponding land (labeled "Ld") surface values (which were previously included).  We have also added additional trends over various latitude bands.

3. The manuscript is very well structured and quite well written, but the heavy use of acronyms and technical identifiers (e.g., SSP3-7.0-lowNTCF, lowAERO3, etc.) makes it harder to read & follow than it needs to be. I would encourage the authors to simplify this wherever possible and then use a consistent, easy to interpret nomenclature throughout. For example, frequently the two scenarios are referred to as strong and weak air quality control, and these are much easier to interpret than SSP3-7.0 andSSP3-7.0-lowNTCF. I would suggest strong and weak air quality control could replaceSSP3-7.0 and SSP3-7.0-lowNTCF everywhere, in particular in figure legends and captions where the reader may not be referring back to the text. Similarly, NTCF mitigation is easier to interpret than SSP3-7.0-lowNTCFSSP3-7.0.

We have removed some of the acronyms.  In particular, we now use strong and weak air quality control, as well as NTCF mitigation.  We also use more straightforward acronyms for the two model subsets, Aer and Aer+O3, as suggested by Reviewer #1.

4. The manuscript cites a lot of "in prep" and "submitted" papers. In most cases, these are cited as part of long lists of other references, so they aren't really needed to make the points. If these are not at least in ACPD by the time of publication, they should not be included in the citation lists (except in cases where they are the only publications available to back-up the point).

All references have been updated.

SPECIFIC COMMENTS

L30: Does the net radiative effect here refer just to OC or to BC+OC? Please rephrase to clarify.

This statement has been clarified.  This is the best estimate of net industrial-era climate forcing by all short-lived species from black-carbon-rich sources.

L59: Is this newer estimate of mortality for all air pollution or outdoor ambient only?  Please rephrase to clarify.

This is for all air pollution.  Clarified.

L90-108: This information would benefit from being summarised in a table listing the scenarios and some of the relevant information (e.g. air quality controls weak/strong,ozone and aerosols high/low, CH4 high/high, etc.) to make it easier for the reader to synthesise.

This information has been moved to this section ("Future Scenarios"), including Figures 1 and 2, which show the global evolution of emission species and the regional trends.  Since only two scenarios are addressed in this manuscript, we only show results from SSP3-7.0 and SSP3-7.0-lowNTCF.

L120-122: I find this a bit confusing. What is the difference between CESM2 and CESM2-WACCM in this case? Is it the aerosol treatment? And if they are basically the same model, is it fair to include them as two separate data points in the multi-model means?

We have removed CESM2 from the analysis.

L141-144: So nitrate aerosol was not included in PM2.5 at all, even for the models that do include it? It would be nice to see how much uncertainty this adds, given nitrate can be an important component of aerosol loading in some regions. I'd suggest adding a version of the PM2.5 figures including nitrate to the supplement, and a brief discussion of the impacts of excluding nitrate either in the main text or in the supplement.

Only one model includes nitrate aerosol data (GFDL-ESM4). Globally (over land only), nitrate decreases by -0.0396 (-0.1165) µg/m^3. These trends are 17 and 20% of the magnitude of the corresponding PM2.5 trends. GFDL-ESM4 also archives ammonium. Globally (over land only), ammonium decreases by -0.0487 (-0.1168) µg/m^3. These trends are 21 and 20% of the magnitude of the corresponding PM2.5 trends. Thus, excluding nitrate and ammonium in GFDL-ESM4 leads to ~40% underestimation of the global PM2.5 trend.

CESM2-WACCM also archives ammonium. Here, however, the global (land) trends are much smaller at -0.00329 and -0.0081 µg/m^3, which leads to ~1% underestimation of the global PM2.5 trend.

This has been added to the revision, as have supplementary figures that show the spatial trend maps for nitrate and ammonium. We have also added a discussion and supplementary figures that compare archived versus estimated PM2.5 trends in 4 models (those 4 that included archived PM2.5).

L156: Is there a reference for these ground-based observations? Or is this the same GASSP observations mentioned above? If the latter, please state explicitly in the text.

This is referring to GASSP. Fixed.

L172-L180: This is confusing when paired with the figure. It is completely legitimate to not include the differences in CH4 pathways for this work, but anyone skimming quickly and focusing on the figures will miss that point. In my opinion, Figure 1 should only show what was used in this work, not scenarios that are not used here. I strongly encourage the authors to remove the SSP3-7.0-lowNTCF (right?) and difference lines from Figure 1. The comparison between the scenarios can be moved to the supplement if the authors feel it is important to include.

The SSP3-7.0-lowNTCF and difference CH4 data have been deleted from Figure 1 (same for Figure 2). We have also removed SSP3-7.0-lowNTCF CO2 from these figures. Both sets of AerChemMIP simulations use the same CO2 and CH4 data, based on SSP3-7.0.

L181-187: Similarly, I don't think this discussion belongs here. It is the first section of the results, yet it is mostly discussing what is not done in this work. I would suggest this could be

removed entirely, or moved to the supplement or to the conclusions as part of a discussion of what future work should be done to build on what the authors have done here.

This discussion has been removed.

Sect. 3.2 and Figs 3-4: Generally speaking, is the changes in atmospheric composition (aerosols and ozone) that are driving the changes in climate. Thus it seems a bit odd to show and discuss the changes in climate variables BEFORE the changes in composition (ozone, PM2.5). I would suggest restructuring such that Fig.3 comes before Fig. 4, with the text order changed to match. (I note this is already the order used in the abstract and conclusions).

Re-ordered according to the reviewer's suggestion.

L204-205: Unless you rename & define the scenarios in the methods as discussed above, please clarify how "under NTCF mitigation" is defined here (I understand that itis the difference between the two scenarios, but that wasn't clear to me on first read).

Scenarios have been defined according to the reviewer's suggestion. SSP3-7.0 is referred to as weak air quality control and SSP3-7.0-lowNTCF is referred to as strong air quality control. Their difference (strong minus weak air quality control) is referred to as NTCF mitigation.

L211-218, L223-228 (and elsewhere): Much is made of the difference between the lowAER and lowAERO3 outcomes. Given that one of these only includes 3 models and the manuscript states explicitly that the difference is not significant, it is not justifiable to be interpreting this as a result. This appears to me to be over-interpretation of noise, and I would suggest this discussion be removed before publication in ACP.

We agree, and this statement has been modified. It is still interesting that similar (i.e., non-significant differences) global mean surface air temperature trends occur in Aer and Aer+O3 models. We acknowledge that this could be due to several factors, but one interpretation is weak surface cooling due to reductions in ozone.

L233: This land-only result appears to be insignificant for 75% of the models (including those that show increases) and so this statement should be removed or qualified.

Sentence has been deleted.

L238: CDD does not show a statistically significant increase in the overall MMM (or in the subset MMM or in any of the individual models bar one) – therefore should be removed from this sentence.

Deleted.

L255-256: Is the land-only warming pattern shown anywhere? Is the land-only warming weaker or stronger than the overall warming? If there is a difference, it would be useful to see an

equivalent figure in the supplement. (And if there is not a difference,it's not clear why this is discussed separately.)

Table 1 shows that the land warming is stronger than that over both land and ocean. This has been clarified. Surface temperature trend patterns are included in the Supplement.

L264: "...forcing and response do not need to occur in the same regions." Can this be explained a bit more?

A sentence has been added.

L269-271: Do I understand Fig 6 bottom panel correctly that models don't agree about this feature? If so that would be worth stating in this discussion

Figure 6 bottom panels show the percentage of models that agree on the sign of the trend. Red colors indicate model agreement on a positive trend; blue colors indicate model agreement on a negative trend. White areas indicate lack of agreement on the sign of the trend. The caption has been clarified. About 70% of the models agree that the North Atlantic cools.

L307: For a discussion of seasonal patterns to make sense, consideration should be given to the different seasonalities of the two hemispheres. Figure 7 should either be separated or at least ordered/demarcated by hemisphere – I'd suggest NH extratropics, tropics, and SH tropics.

Figure 7 shows seasonal trends for each of our 12 world regions. Thus, this figure is already broken down into regional demarcations consistent with seasonality. Nonetheless, we have also added trends for several latitude bands, including those requested.

L370: Why is one model listed explicitly when all (including that model) are available from the same location? Also please spell out ESGF here and provide a link or doi.

Reference to GFDL deleted. ESGF spelled out, and a link is provided.

Figs 2, 5,7: regional legend labels on x-axis are impossible to read because they are so small. Perhaps give each region a number instead? Or include some other sort of key to make this clearer?

We have modified the x-axis on these figures. A key is now used.

Fig 3 caption (and elsewhere): Does "hottest day" refer to "surface temperature on hottest day"? Similarly for wettest day? Please clarify somewhere.

Extreme weather indices are defined in the Methodology section.

We also analyze climate extremes including the hottest day (monthly maximum value of daily maximum surface temperature), wettest day (monthly maximum 1-day surface precipitation) and consecutive dry days (CDD), defined as the maximum annual number of consecutive days with

surface precipitation less than 1 mm/day.   We focus on these three extreme indices since they are frequently used metrics for temperature and precipitation extremes.  Prior observational analyses have shown significant increases (decreases) in the hottest and wettest day (CDD) over the latter half of the 20th century (Donat et al, 2013a,b).  Climate extremes are based on daily data, and are calculated at each grid box and then spatially averaged.

Fig 3 caption: It seems the thin coloured lines show the trends for the individual models, but this has not been explicitly stated in the caption. Please update caption to clarify.

Thin (and non-black lines) show individual model mean trends.  Line colors are denoted by the legend.  We have also added this to the caption.  Same for Figure 4.

Figs 3, 4, 5: why are different units used for the trends in the precip variables (mm/dayvs. %) in the global and regional trend figures? Same question for PM2.5 and O3. Can these be standardised to more easily compare?

Sure.  We no longer use percent changes.

Fig 6d-f: these plots are not currently discussed in the text and therefore should per-haps move to the supplement (or be mentioned in the text)

A sentence pertaining to these panels has been added.

Table 1: I found this table hard to understand while trying to refer to it while reading the text. A few suggestions to improve the clarity. (1) Add lowAER and lowAERO3 identifiers above the list of relevant models in each sub-section so it's easy to see which group is which. (2) If text and figures are re-ordered as suggested above, move PM2.5 and O3 columns to be left-most, followed by the climate variables. (3) Move the three "MMM total" lines either to a separate part of the table or (preferably) to a new table altogether as the numbers aren't comparable to the lines above/below which makes it difficult to interpret (and already a lot to interpret in the table!). (4) For the lowAER models' O3 response, replace 0.0 with n/a since these values are not included in the Overall MMM calculation (as is, looks like the overall will be an average of thelowAERO3 values and three zeros).

Made all suggested modifications to Table 1.

TECHNICAL COMMENTS

L139: "were are" –> "we are"

L357: "complex" –> "complexity"

Fig 2 caption: "astriks" –> "asterisks"

All have been fixed.

---

## Author Response (AR2)

**Response to Reviewers**

We thank Reviewer #1 and the Editor for re-evaluating our manuscript. We list below (in blue) how we have addressed the remaining concerns.

**Editor Comments**

In addition to the reviewers' comments, I suggest some further improvements.

The abstract could be a bit clearer to serve as a stand-alone text.

For example:
L7 I miss the assertion that the choice for SSP3 without and without air pollution policies, leads to a 'maximum' effect, and as a consequence using other SSPs would likely give lower impacts.

We have added the following sentence to the abstract:

As SSP3-7.0 lacks climate policy and has the highest levels of NTCFs, our results (e.g., surface warming) represent an upper bound.

Abstract lines 9-15, the quantification of ozone/pm2.5 changes; as well as the temperature, precipitation: please clarify whether these numbers apply to the changes in the SSP3 baseline versus SSP-3 NTCF scenario by 2050, or averaged over the simulation period. Would it be possible to add uncertainty intervals?

The quoted changes represent the total change from 2015-2055. This has been clarified in the abstract. We have also added uncertainty (95% confidence intervals) to the quoted changes in the abstract.

Clarify also how the choice of these two scenarios provides information on "policies that optimally address both climate change and air quality", as this is clearly not the case for the SSP3 vs SSP3-NTCF scenario. SSP3 rather expresses the absence of climate policy.

Yes, the two scenarios used here are not meant to explicitly illustrate how to optimally address climate change and air quality. However, they do show the impacts of reducing NTCFs (aerosols and ozone). Although this improves air quality, it also leads to continued surface warming. Thus, additional emission reductions need to occur, as described in the text, including methane and $CO_2$. As mentioned below, additional simulations are being conducted that include the methane reductions in SSP3-7.0-lowNTCF. Future analysis of these simulations will provide additional information on the climate and air quality effects of all NTCFs, including methane.

We have replaced this sentence with the following:

It is important to understand how future environmental policies will impact both climate change and air pollution.

L21 Suggest to introduce also the term SLCF, along with NTCF, as many will know this term better than NTCF. IPCC considers the two synonymous.

We have added the SLCF nomenclature near L21.

L50 Clarify whether the use of the word likely (here and following sentences) implies a quantified uncertainty, as in IPCC uncertainty language? This is in particular of importance, as IPCC will probably use this paper in its forthcoming assessment.

We have replaced instances of "likely" with a suitable synonym.

L61 The risk factor is for premature death and mainly due to non-communicable diseases.

This sentence has been added near L61.

L116 Some more elaboration on the issue of 3 realisation vs 1 realisation for these transient simulations would be useful. What is the range within single models of the results, and what does it imply for uncertainties from single realisations. Is this information used, or could this information be used for improving error estimates of the multi model average?

There are several sources of uncertainty. For the context of this comment, the most important sources of uncertainty are model differences, as well as internal climate variability. Our quoted uncertainties include both. However, if we had three realizations for each model, the role of internal climate variability could be better isolated. Although three realizations are probably not enough to truly quantify this source of uncertainty. This is the goal of the large ensemble projects currently being conducted. Similarly, additional models would also allow improved quantification of the uncertainty due to model differences.

For a given variable we analyze, the uncertainty across realizations for models with multiple runs is comparable to the uncertainty across models. For example, in terms of the total global surface temperature change, we quote a multi-model mean 95% confidence interval of 0.12 K. Models with multiple realizations yield corresponding values of 0.02, 0.08, 0.10, 0.11, 0.15, 0.22, which yields an average of 0.11 K. Similarly, in terms of the total global land PM2.5 change, we quote a multi-model mean 95% confidence interval of 0.32 $\mu g/m_3$. Models with multiple realizations yield corresponding values of 0.08, 0.09, 0.16, 0.25, 0.29, which yields an average of 0.17 $\mu g/m_3$.

If all models and realizations are used, our uncertainty estimates are reduced (due, in part, to more data). For example, the 95% confidence interval for total global surface temperature change decreases from the quoted 0.12 to 0.07 K. The 95% confidence interval for the total global land PM2.5 change decreases from 0.32 to 0.24 $\mu g/m_3$.

Some of this information has been added to the revision, under "Model Data and Methodology".

L190 explain the rationale of not including methane the experiments.

Methane is not included here so that the response of aerosols and ozone alone is isolated. Modeling groups are currently conducting (or will be shortly) similar SSP3-7.0-lowNTCF experiments that are analogous to those presented here, but also include the methane reductions (so called SSP3-7.0-lowNTCFCH4 experiments). This information has been added to the revision.

L191 I didn't understand why it is difficult to reduce air pollutant- while not changing CO2. Isn't that the case for e.g. any filter, scrubber, catalysts on top of existing technology. Or is the idea that technology changes are needed to achieve further air pollutant emission reductions, and those technologies are probably also delivering on GHG gas emission. The terminology 'keep CO2 fixed' is confusing, you keep the scenario assumptions constant, but the emissions change according to SSP3.

We've deleted this sentence. Our main point, which is expressed just prior to this sentence, is that SSP3-7.0-lowNTCF represents an idealized scenario (two scenarios are essentially cobbled together) as it replaces SSP3's lack of stringent air quality policy with SSP1's strict air pollutant legislation and technological progress.

L402 Conclusions. It seems that the omission of aerosol nitrate from 3 of 4 models, may lead to a substantial low bias in the estimations of aerosol and resulting effects. It would be appropriate to mention in this in the conclusions.

A sentence has been added to the conclusions:

We also reiterate that few models include nitrate aerosol, which implies an underestimation bias in the climate responses shown here.

**Reviewer #1**

Comments to the revised version.

The authors have included a detailed section on the statistics, which clarifies many of my questions. Overall, I think the paper is in good shape and can be published subject to two minor, though I think important, corrections.

1) Abstract and Introduction on uncertainty

line 3-4 "The climate impacts ... remain uncertain".

Thanks for the clarification. Note that the wording "remain uncertain" might be perceived as "not

known at all". I still would like to have this sentences clarified a bit more. e.g. "Our results show that trends in climate impacts of future policies that address the abundance of NTCFs and air quality can be assessed, but only with large remaining uncertainties." and you might find a better wording which includes (1) to clarify if this sentence is based on your results and (2) clarify if the a tendency can be deduced within a range of uncertainty or whether you think that everything is just uncertain.

Note that you show very detailed results afterwards (0.15 mm/day), how can that be done if the result is uncertain?

Lines 3-4 pertain to prior studies.  As discussed in the introduction, prior assessments of the impact of NTCF mitigation on air quality and climate have been limited.   This is related to the idealized nature of some prior studies, simplified treatment of aerosols and chemically reactive gases, as well as a lack of sufficiently large number of models to quantify model diversity and robust responses.

We have clarified what we meant by "uncertain" in L3-4.  We have modified to:

Prior assessments of the impact of NTCF mitigation on air quality and climate have been limited. This is related to the idealized nature of some prior studies, simplified treatment of aerosols and chemically reactive gases, as well as a lack of sufficiently large number of models to quantify model diversity and robust responses.

The quoted numbers in the abstract (e.g., 0.15 mm/day) are from this study.  We have added the uncertainty (95% confidence intervals) associated with these changes.

line 82 "Despite the rich literature, the impact of NTCF mitigation on climate and air quality remains uncertain. "

Also here, I would appreciate a more detailed description of what you mean with "uncertain". Perhaps: Despite the rich literature, a robust assessment of the impact of specific NTCF mitigation measure on climate and air quality can hardly be achieved.

Otherwise the sentence can be interpreted, e.g., as if 2 state-of-the-art models give for the same mitigation option always answers, which even differ in sign.

We have adopted the reviewer's suggestion for L82, which now reads:

Despite the rich literature, a robust assessment of the impact of specific NTCF mitigation measures on climate and air quality has been difficult to achieve.

2) Motivation for indicators
I appreciate the clarification in Section 2.2! And yes the use of these metrics are sort of motivated. But I also strongly suggest to add a sentence, e.g. at line 92 that clearly states what indicators you are using. We use mean surface temperature, precipitation, hottest and wettest

days as indicators for climate change and surface O3 and PM2.5 for air quality as those are the most commonly used metrics.

Near L92, we have added the following sentence:

[revised manuscript text omitted]